# Modular System for Shelves and Coasts (MOSSCO v1.0) – a flexible and multi-component framework for coupled coastal ocean ecosystem modelling

Carsten Lemmen[1], Richard Hofmeister[1,4], Knut Klingbeil[2,5], M. Hassan Nasermoaddeli[3,6], Onur Kerimoglu[1], Hans Burchard[2], Frank Kösters[3], and Kai W. Wirtz[1]

[1]Institute of Coastal Research, Helmholtz-Zentrum Geesthacht Zentrum für Material- und Küstenforschung, 21502 Geesthacht, Germany
[2]Department of Physical Oceanography and Instrumentation, Leibniz-Institute for Baltic Sea Research, 18119 Rostock-Warnemünde, Germany
[3]Section Estuary Systems I, Bundesanstalt für Wasserbau, 22559 Hamburg, Germany
[4]Institute for Hydrobiology and Fisheries Science, Universität Hamburg, 22767 Hamburg, Germany
[5]now at Department of Mathematics, University of Hamburg, 20146 Hamburg, Germany
[6]now at Landesbetrieb Straßen, Brücken und Gewässer, Freie und Hansestadt Hamburg, 20097 Hamburg, Germany

*Correspondence to:* C. Lemmen
(carsten.lemmen@hzg.de)

**Abstract.** Shelf and coastal sea processes extend from the atmosphere through the water column and into the sea bed. These processes reflect intimate interactions between physical, chemical, and biological states at multiple scales. As a consequence, coastal system modelling requires a high and flexible degree of process and domain integration; this has so far hardly been achieved by current model systems. The lack of modularity and flexibility in integrated model hinders the exchange of data and model components and has historically imposed supremacy of specific physical driver models. We here present the Modular System for Shelves and Coasts (MOSSCO, http://www.mossco.de), a novel domain and process coupling system tailored—but not limited—to the coupling challenges of and applications in the coastal ocean. MOSSCO builds on the Earth System Modeling Framework (ESMF) and on the Framework for Aquatic Biogeochemical Models (FABM). It goes beyond existing technologies by creating a unique level of modularity in both domain and process coupling including a clear separation of component and basic model interfaces, flexible scheduling of several tens of models and facilitation of iterative development at the lab, the station, and the coastal ocean scale. MOSSCO is rich in metadata and its concepts are applicable also outside the coastal domain. For coastal modelling, it contains dozens of example coupling configurations and tested setups for coupled applications. Thus, MOSSCO addresses the technology needs of a growing marine coastal Earth System community that encompasses very different disciplines, numerical tools, and research questions.

## 1 Introduction

Environmental science and management consider ecosystems as their primary subject, i.e. those systems where the organismic world is fundamentally linked to the physical system surrounding it; there exist neither unequivocally defined spatial nor pro-

cessual boundaries between the components of an ecosystem (Tansley, 1935). Consequently, holistic approaches to ecological research (Margalef, 1963), to biogeochemistry (Vernadsky, 1998, originally 1926) and to environmental science in general (Lovelock and Margulis, 1974) have been called for.

The need for systems approaches is perhaps most apparent in coastal research. Shelf and coastal seas are described by components from different spatial domains: atmosphere, ocean, soil; and they are driven by a manifold of interlinked processes: biological, ecological, physical, geomorphological, amongst others. Crossing these domain and process boundaries, the dynamics of suspended sediment particles (SPM, see Table 2 for abbreviations) and of living particles, or the interaction between water attenuation and phytoplankton growth, for example, are both scientifically challenging and relevant for the ecological state of the coastal system (e.g., Shang et al., 2014; Maerz et al., 2011; Azhikodan and Yokoyama, 2016).

For historical and practical reasons, the representation of the coastal ecosystem in numerical models has been far from holistic. Most often, ecological and biogeochemical processes are described in modules that are tightly coupled to one or a few hydrodynamic models. For example, the Pelagic Interactions Scheme for Carbon and Ecosystem Studies (PISCES, Aumont et al., 2015) has been integrated into the Nucleus for European Modeling of the Ocean (NEMO, Van Pham et al., 2014) and the Regional Ocean Modeling system (ROMS, Jaffrés, 2011). Or, the Biogeochemical Flux Model (BFM) has been integrated in the Massachusetts Institute of Technology Global Circulation Model (MITgcm) (Cossarini et al., 2017) and ROMS. These tight couplings not only exclude important processes at the edges of or beyond the pelagic domain, they also lack flexibility to exchange or to test different process descriptions.

To stimulate the development, application and interaction of ecological and biogeochemical models independently of a single host hydrodynamic model, Bruggeman and Bolding (2014) presented the Framework for Aquatic Biogeochemical Models (FABM), which serves as an intermediate layer between the biogeochemical zero-dimensional process models and the three-dimensional geophysical environment models. FABM has been implemented in the Modular Ocean Model (MOM, Bruggeman and Bolding, 2014), NEMO, the Finite Volume Coastal Ocean Model (FVCOM, Cazenave et al., 2016), or the General Estuarine Transport Model (GETM, Kerimoglu et al., 2017). With more than 20 biogeochemical and ecological models included, FABM has enabled marine ecosystem researchers to describe the system's many aquatic processes.

The process-oriented modularity realized within FABM, however, lacks the means to describe cross-domain linkages. Historically rooted in atmosphere–ocean circulation models (Manabe, 1969), the coupling of earth domains is the standard concept in Earth System Models (ESM). Domain coupling is also a major challenge in coastal modelling and has been used, for example, in the Coupled Ocean-Atmosphere-Wave-Sediment Transport (COAWST, Warner et al., 2010) system. COAWST comprises the Regional Ocean Modeling System (ROMS) with a tightly coupled sediment transport model, the Advanced Research Weather Research and Forecasting (WRF) atmospheric model, and the Simulating Waves Nearshore (SWAN) wave model. Each of the components in domain coupling is usually a self-sufficient model that is run in a special "coupled mode". Interfacing to other components is done via coupling infrastructure, such as the Flexible Modeling System (FMS, Dunne et al., 2012), the Model Coupling Toolkit (MCT, Warner et al., 2008) and/or the Ocean Atmosphere Sea Ice Soil (OASIS) coupler (Craig et al., 2017), or the Earth System Modeling Framework (ESMF, Theurich et al. 2016, see, e.g., Jagers 2010 for an intercomparison of coupling technologies). Recently, Pelupessy et al. (2017) introduced the Oceanographic Multipurpose Software

Environment (OMUSE) and demonstrated nested ocean and ocean-wave domain couplings. Their intention is to provide a high-level user interface and infrastructure for coupling existing and new oceanographic models whose spatial representations differ greatly, in particular between Lagrangian and Eulerian type representations. The Community Surface Dynamics Modeling System (CSDMS, Peckham et al., 2013) even allows to couple models implemented in many different languages, as long as all of these describe their capabilities in basic model interface (BMI, Peckham et al., 2013) descriptions. Typically though, only three to five domain components are coupled through one of the above technologies (Alexander and Easterbrook, 2015).

The differentiation between domain and process coupling is not a technical necessity: A typical domain coupling software like ESMF can also be used to couple processes: with the Modeling, Analysis and Prediction Layer (MAPL, Suarez et al., 2007), the Goddard Earth Observing System version 5 (GEOS-5) encompasses 39 process models coupled hierarchically through ESMF; development of these modules, however, is strictly regulated within the developing laboratory. Vice versa, a typical process coupling infrastructure like the Modular Earth Submodel System (MESSy, Jöckel et al., 2005), which initially linked mostly atmospheric processes, has been generalized to support linking at a user-chosen granularity irrespective of the process versus domain dichotomy (e.g., Kerkweg and Jöckel, 2012).

Up to now, there was no coastal modelling environment that enables a modular and flexible process (model) integration and cross-domains coupling at the same time, and that is open to a larger community of independent biogeochemical and ecological scientists. The underlying long-term goal for increasingly holistic model systems conflicts with the evolving and diverse research needs of individual scientists or research groups to address very specific problems; it remains difficult to link up-to-date research that is delivered at the (local) process scale to the Earth System scale. Thus we here present the Modular System for Shelves and Coasts (MOSSCO, www.mossco.de), a novel system for domain and process coupling that is tailored— but not limited—to the coupling challenges of and applications in the coastal ocean. This new system builds on the flexibility of FABM and on the infrastructure provided by ESMF with its cross-domain and many-component hierarchical capability. We here present the major design ideas of MOSSCO and briefly demonstrate its usability in a series of coastal applications.

## 2 MOSSCO concepts

The modularity and coupling concepts proposed in this paper describe a novel software system that addresses the needs of researchers who want to make maximum use of their existing knowledge in a specific field (e.g., geomorphology or marine ecology) but wish to conduct integrative research in a wider and flexible context. In strengthening modularity *sensu* independence of specific physical drivers, the new concept should, in addition to addressing the problems listed above, support (1) liaisons between traditionally separated modelling communities (e.g., coastal engineers, physical oceanographers and biologists), (2) inter-comparison studies of, e.g., physical, geological, and biological modules, and (3) up-scaling studies where models developed at the laboratory scale in a non-dimensional context are applied to regional, global and Earth System scales.

The design of MOSSCO is application-oriented and driven by the demands for enabling and improving integrated regional coastal modelling. It is targeted towards building coupled systems that support decision making for local policies implementing

the European Union Water Framework Directive (WFD) and Marine Strategic Planning Directive (MSPD). From a design point of view we envisioned a system that is foremost flexible and equitable.

**Flexibility** means that the system itself is able to deal on the one hand with a diverse small or large constellation of coupled model components and on the other hand with different orders of magnitude of spatial and temporal resolutions; it is able to deal equally well with zero-, one-, two- and three-dimensional representations of the coastal system. Flexibility implies the capability to encapsulate also existing legacy models to create one or more different "ecosystems" of models. This feature should allow seamless replacement of individual model components, which is an important procedure in the continual development of integrated systems. Flexibly replacing components finally creates a test-bed for model intercomparison studies.

**Equitability** means that all models in the coupled framework are treated as equally important, and that none is more important than any other. This principle dissolves the primacy of the hydrodynamic or atmospheric models as the central hub in a coupled system. Also, data components are as important as process components or model output; any de facto difference in model importance should be grounded on the research question, and not on technological legacy. As complexity grows by coupling more and more models, this equitability also demands that experts in one particular model can rely on the functionality of other components in the system without having to be an expert in those models, as well.

The equitability design extends to participation: contributions to the development of components or the coupling framework itself is allowed and encouraged. Anyone can use and modify the coupled framework or parts of both in a legal sense by open source licensing, and in an accessibility sense through template codes and extensive documentation.

### 2.1 Wrapping legacy models – first steps in `PARSE`

As MOSSCO is built around the ESMF hierarchy of components, any existing code that can be wrapped in an ESMF component can be a component in MOSSCO, too. The ESMF user guide (ESMF Joint Specification Team, 2013) suggests a best practice method `PARSE` to achieve this componentization of a legacy code.

**P** repare the user code by splitting it into three phases that initialize, run and finalize a model;

**A** dapt the model's data structures by wrapping them in ESMF infrastructure like states and fields;

**R** egister the user's initialize, run, and finalize routines through ESMF;

**S** chedule data exchange between components;

**E** xecute a user application by calling it from an ESMF driver.

This `PARSE` concept allows a smooth transition from a legacy model to an ESMF component. In this concept, the first three steps have to be performed on the model side, and the latter two on the framework side and have been taken care of by the

MOSSCO coupling layer. The *preparation* of the code is independent of the use of ESMF and provides the basic coupling ability—or "coupleability"—of the model; many existing models already implement this separation into initialize, run, and finalize phases, either structurally, or more formally by implementing a BMI. For the run phase, it is mandatory to refer to a single model timestep and not to the entire run loop.

The *adaption* of a model's internal structures to ESMF consists of technically wrapping data into ESMF communication objects, and in providing sufficient metadata for communication. Among these are grid definition and decomposition, units and semantics of data, optimally following a metadata scheme like the widespread Climate and Forecast (CF, Eaton et al., 2011) or the more bottom-up Community Surface Dynamics Modeling System (CSDMS, following a scheme like object + operation + quantity, Peckham 2014). Both are currently being included in the emerging Geoscience Standard Names Ontology (GSN, geoscienceontology.org).

ESMF provides the interfaces for models written in either the Fortran or C programming languages; data arrays are bundled together with related metadata in ESMF field objects. All field objects from components are then bundled into exported and imported ESMF state objects to be passed between components. As a third step, the ESMF *registration* facility needs to be added to a user model; this step is achieved by using template code from any one of the examples or tutorials provided with ESMF. The second and third step (*adapt* and *register*) are typical tasks of what Peckham et al. (2013) refers to as a component model interface (CMI); it is very similar between models (and thus easily accessible from template code) and targets the interface of a specific coupling framework.

MOSSCO contains CMIs for ESMF in all of its provided components (Fig. 1). The current naming scheme follows the CF convention for standard names except for quantities that are not defined by CF; these names (often from biological processes) are modelled onto existing CF standard names as much as possible. MOSSCO also allows the specification within other naming schemes and includes a name matching algorithm to mediate between different schemes. For future development, adoption of the GSN ontology is foreseen.

## 2.2 Scheduling in a coupled system – the "S" in `PARSE`

MOSSCO adds onto ESMF a scheduling system (corresponding to the fourth step in `PARSE`) that calls the different phases of participating coupled models. The coupling time step duration of this new scheduler relies on the ESMF concept of alarms and a user specification of pairwise coupling intervals between models. The scheduler minimizes calls to participating models by flexibly adjusting time step duration to the greatest common denominator of coupling intervals pertinent to each coupled model. Upon reading the user's coupling specification, (i) models are initialized in random order but with consideration of special initialization dependencies set by the user; (ii) a list of alarm clocks is generated that considers all pairwise couplings a model is involved in; (iii) special couplers associated with a pairwise coupling are executed; (iv) the scheduler then tells each model to run until that model reaches its next alarm time; (v) advancing of the scheduler to the minimum next alarm time repeats until the end of the simulation.

The MOSSCO scheduler allows for both sequential and for concurrent coupling of model components, or a hybrid coupling mode. In the concurrent mode, components run at the same time on different compute resources; in the sequential mode,

components are executed one after another on the same set of compute resources. Recently, Balaji et al. (2016) demonstrated how a hybrid coupling mode and fine granularity could be used to increase the performance of a system that consists of both highly scalable and less scalable components: In their system, an ocean and an atmosphere component run concurrently; within the atmospheric component, the radiation code is executed concurrently to a composite component containing that encompasses in a sequential coupling of all non-radiative atmospheric processes.

For both concurrent and sequential modes, coupling between components is explicit: the MOSSCO scheduler runs the connectors and mediator components that exchange the data before the components are run. For sequential mode, the coupling configuration allows also a memory efficient scheme where consecutive components operate on shared data that always reflects the most recently calculated data from the previous component (Fig. 2, see also Sect. 3.3.1); such sequential coupling on shared data potentially introduces mass imbalances.

Users specify the coupling in a text format using YAML (short for YAML Ain't Markup Language, http://yaml.org) notation, a human reader friendly data serialization standard. The item `coupling` contains a list of `components` items that itself contain a list of coupled components; in the simple example (Fig. 3a) two components named "A" and "B" are coupled. By default, these components are coupled in sequential mode with the default connector sharing their data; the execution order of "A" and "B" is not specified. In a more elaborate example (Fig. 3b), the order of components in the scheduler is specified in the `dependency` section, indicating a run sequence of first A, then B, last C; all components run on the same set of compute resources in (default) sequential coupling mode.

The `instances` section declares that the component named "C" is an instance (or copy) of "A"; this makes it possible to reuse components multiple times in (possibly different) configurations. Typically, data reader or writer components are instantiated from a generic input/output component to access different files for model input and output. Multiple couplings between the three components "A", "B", and "C" are present with coupling `intervals` that lead to scheduling of coupling events according to Fig. 2. Between "A" and "C" a special coupler "D" handles the data exchange instead of the default connector.

## 2.3 Deployment of the coupled system – the "E" in **PARSE**

MOSSCO provides a Python-based generator that dynamically creates an ESMF driver component in a star topology that then acts as the scheduler for the coupled system. This generator reads the specification of pairwise couplings (Fig. 3) and generates a Fortran source file that represents the scheduler component. The generator takes care of compilation dependencies of the coupled models, and of coupling dependencies, such as grid inheritance; in addition to the basic init–run–finalize BMI scheme, it also honors multi-phase initialization (as in the National Unified Operational Prediction Capability, NUOPC, ESMF extension) and a restart phase. The generated code structurally and functionally resembles a NUOPC driver, but it does not require implementation of the NUOPC extension, which is currently restricted to handling only structured grid based submodels.

A MOSSCO command line utility provides a user-friendly interface to generating the scheduler, (re-)compiling all source codes into an executable and submitting the executable to a multi-processor system, including different high-performance computing (HPC) queueing implementations; this is the fifth step in `PARSE`. By designing this command line utility and

automatic scheduler component creation based on the simple YAML textual coupling specification, MOSSCO provides a fast way to reconfigure, rearrange, extend or reduce coupled systems very quickly, in contrast to more elaborate graphical coupling tools such as the CUPID Eclipse interface (Dunlap, 2013, only for NUOPC).

MOSSCO has been successfully deployed at several national HPC centers, such as the Norddeutsche Verbund für Hoch- und Höchstleistungsrechnen (HLRN), the German Climate Computing Center (DKRZ), or the Jülich Supercomputing Centre (JSC); equally, MOSSCO is currently functioning on a multitude of Linux and macOS laptops, desktops and multiprocessor workstations using the same MOSSCO (bash-based) command line utility on all platforms.

The MOSSCO coupling layer is coded in Fortran while most of the supporting structure is coded in Python and partially in Bash shell syntax. The system requirements are a Fortran 2003 compliant compiler, the CMake build system, the Git distributed version control system, Python with YAML support (version 2.6 or greater), a Network Common Data Form (NetCDF, Rew and Davis, 1990) library, and ESMF (version 7 or greater). For parallel applications, a Message Passing library (e.g., OpenMPI) is required. Many HPC centers have toolchains available that already meet all of these requirements. For an individual user installation, all requirements can be taken care of with one of the package managers distributed with the operating system, except for the installation of ESMF, which needs to be manually installed; MOSSCO provides a semiautomated tool for helping in this installation of ESMF. The steps to get MOSSCO running quickly on any suitable computer system are outlined in Fig. 3c. These instructions should get a reader started on carrying out first simulations with a coupled system by typing a dozen lines of code, provided that all requirements are met.

## 3  MOSSCO components and utilities

Driven by user needs, MOSSCO currently entails utilities for I/O, an extensive model library, and coupling functionalities (Fig. 4 and Table 1). As a utility layer on top of ESMF, MOSSCO also extends the Application Programming Interface (API) of ESMF by providing convenience methods to facilitate the handling of time, metadata (attributes), configuration, and to unify the provisioning and transfer of scientific data across the coupling framework. The use of this utility layer is not mandatory; any ESMF based component can be coupled to the MOSSCO provided components without using this utility layer.

One of the major design principles of MOSSCO is seamless deployment from zero-dimensional to three-dimensional spatial representations, while maintaining the coupling configuration to the maximum extent possible. This design principle builds on the dimensional-independency concept of FABM achieved by local description of processes (often referred to as a box model), where the dimensionality is defined by the hydrodynamic model to which FABM is coupled; MOSSCO generalizes this concept to enable especially the developers of new biological and chemical models to scale up from a box-model (zero-dimensional) to a water-column (one-dimensional), sediment plate or a vertically resolved transect (two-dimensional), and a full atmosphere or ocean (three-dimensional) setup. As a concrete example, the novel Model for Adaptive Ecosystems in Coastal Seas (MAECS, Wirtz and Kerimoglu, 2016; Kerimoglu et al., 2017) has been developed iterating between an application for the lab (zero-dimensional) scale and the three-dimensional regional coastal ocean scale.

All utility functions and components, especially the model-independent I/O facilities from MOSSCO, are able to handle data of any spatial dimension. Components that do not define their own spatial representation as a grid or mesh are able to inherit the complete spatial information from a coupled component that provides such a grid: Usually (but not necessarily) biological and chemical models inherit the spatial configuration from a hydrodynamic model; equally well, this information can be obtained from data in standardized grid description formats like Gridspec (Balaji et al., 2007) or the Spherical Coordinate Remapping and Interpolation Package (SCRIP, Jones, 1999). Grid inheritance is specified as a dependency in the coupling specification.

### 3.1 Model library: Basic Model Interfaces for scientific model components

The model library (right branch in Fig. 4) includes new models (e.g., for filter feeders and surface waves) and wrappers to legacy models and frameworks such as FABM or GETM. Some of these wrappers are under development, among them the Hamburg Shelf Ocean Model (HamSOM, Harms, 1997) and a Lagrangian particle tracer model. Here, we briefly document the model collection in particular with respect to their preparation and functioning within the new coupling context.

#### 3.1.1 Pelagic ecosystem component

The pelagic ecosystem component (`fabm_pelagic_component`) collects (mostly biological) process models for aquatic systems. This component makes use of the Framework for Aquatic Biogeochemical Models (Bruggeman and Bolding, 2014). FABM is a coupling layer to a multitude of biogeochemical models which provide the source-minus-sink terms for variables, their vertical local movement (e.g., due to sinking or active mobility), and diagnostic data. Each model variable is equipped with metadata, which is transferred by the ecosystem component into ESMF field names and attributes. Similarly, the forcing required by the biogeochemical models is communicated within the framework and linked to FABM. The pelagic ecosystem component includes a numerical integrator for the boundary fluxes and local state variable dynamics. Advective and diffusive transport are not part of this component but are left to the hydrodynamical model through the `transport_connector` (Sect. 3.3.2). The close connection between transport and the pelagic ecosystem requires that the spatial representation of the FABM state variables is inherited from the hydrodynamic model component that performs the transport calculations.

Many well known biogeochemical process models have been coded in the FABM standard by various institutes, such as the European Regional Seas Ecosystem Model (ERSEM, Butenschön et al., 2016), ERGOM (Hinners et al., 2015), PCLake (Hu et al., 2016) and the Bottom RedOx Model (BROM, Yakushev et al., 2017). All pelagic biogeochemical models complying with the FABM standard can equally be used in MOSSCO, while retaining their functionality.

#### 3.1.2 Sediment/soil component

The sediment component `fabm_sediment_component` hosts (mostly biogeochemical) process descriptions for aquatic soils. To allow efficient coupling to a pelagic ecosystem, the sediment component inherits a horizontal grid or mesh from the coupled system and adds its own vertical coordinate, a number of layers of horizontally equal height for the upper soil (typical domain heights range from 10–50 cm). State variables within the sediment are defined through the FABM framework

within the 3D grid or mesh in the sediment. As in the pelagic ecosystem component, the state variables, metadata, diagnostics and forcings are communicated via the ESMF framework to the coupled system. The sediment component is the numerical integrator for the tracer dynamics within each sediment column in the horizontal grid or mesh, including diffusive transport, driven by molecular diffusion of the nutrients or bioturbative mixing. Additionally, a 3D variable porosity defines the fraction of pore water as part of the bulk sediment, while all state variables are measured per volume pore water in each cell. FABM's infrastructure of state variable properties is used to label the new boolean property `particulate` in FABM models to define

whether a state variable belongs to the solid phase within the domain. A typical model used in applications of the sediment component is the biogeochemical model of the Ocean Margin Exchange Experiment OMEXDIA (Soetaert et al., 1996); a version of this model with added phosphorous cycle is contained in the FABM model library as OMEXDIA_P (Hofmeister et al., 2014).

### 3.1.3    1D Hydrodynamics: General Ocean Turbulence Model (GOTM))

The General Ocean Turbulence Model (GOTM, Burchard et al., 1999, 2006) is a one-dimensional water column model for hydrodynamic and thermodynamic processes related to vertical mixing. MOSSCO provides a component for GOTM and a component hierarchy that considers a coupled GOTM with internally coupled FABM within one component (`gotm_fabm_component`), as many existing available model setups rely on the direct coupling of FABM to GOTM. This way, the modularization—taking a coupled GOTM/FABM apart and recoupling it through the MOSSCO infrastructure, can be verified; the encapsulation of

GOTM is implemented in the `gotm_component`.

### 3.1.4    3D Hydrodynamics: General Estuarine Transport Model (GETM)

MOSSCO provides an interface to the 3D coastal ocean model GETM (Burchard and Bolding, 2002). GETM solves the Navier-Stokes Equations under Boussinesq approximation, optionally including the non-hydrostatic pressure contribution (Klingbeil and Burchard, 2013). A direct interface to GOTM (see Section 3.1.3) provides state-of-the-art turbulence closure in the vertical.

GETM supports horizontally curvilinear and vertically adaptive meshes (Hofmeister et al., 2010; Gräwe et al., 2015). The interface to GETM is provided by the `getm_component`; any model coupled to GETM via the transport component can have its state variables conservatively transported by GETM (see Section 3.3.2). In the component, the GETM-created spatial topology is made available as an ESMF grid object; typically this grid and subdomain decomposition is communicated to the coupled system where the spatial and parallelization information is inherited by other components.

**3.1.5    Model components for erosion, sedimentation, and their biological alteration**

The erosion/sedimentation routines of the Deltares Delft3D model (EROSED, van Rijn, 2007) were encapsulated in a MOSSCO component. EROSED uses a Partheniades–Krone equation (Partheniades, 1965) for calculating the net sediment flux of cohesive sediment at the water–sediment interface for multiple SPM size classes. MOSSCO's BMI for uses the current version of EROSED maintained by Deltares; it isolates with the help of subsidiary infrastructure the original code from the deeply

intertwined dependencies in the Delft3D system. The `erosed_component` can provide its own spatial representation as a structured grid or unstructured mesh; it can also inherit the spatial information from a coupled component. The functionality of the erosion/sedimentation component is described in more detail by Nasermoaddeli et al. (2014).

Flow and sediment transport can be affected by the presence of benthic organisms in many ways. Protrusion of benthic animals and macrophytes in the boundary layer changes the bed roughness and thus the bed shear stress and consequently the sediment transport. The erodibility of sediment can be modified by the mucus produced by benthic organisms; the erodibility of the upper bed sediment can be altered by bioturbation generated by macrofauna (de Deckere et al., 2001). In the `benthos_component`, these biological effects of microphytobenthos and of benthic macrofauna on sediment erodibility and critical bed shear stress are parameterized and provided to other coupled components (e.g. the erosion/sedimentation component) as additional erodibility and critical shear stress factors. The benthos effect model is described in detail by Nasermoaddeli et al. (2017).

### 3.1.6 Filter feeding model

The `filtration_component` describes the instantaneous filtration by suspension feeders within the water column. This biological filtration model follows Bayne et al. (1993) and describes the filtration rate as a function of food supply; it can be adapted to different species of filter feeders and was recently applied to describing the ecosystem effect of blue mussels on offshore wind farms as the `filtration_component` of MOSSCO (Slavik et al., "The large scale impact of offshore windfarm structures on pelagic primary production in the southern North Sea", manuscript submitted to Hydrobiologia). The filtration model uses an arbitrary chemical species or compound, say phytoplankton carbon as the "currency" for processing. The amount of ambient phytoplankton carbon concentration is sensed by the model organisms and it is filtered along with the other nutrients (in stoichiometric proportion) out of the environment, creating a sink term for subsequent numerical integration in the pelagic ecological model.

### 3.1.7 Wind waves

A simple wind wave model is part of the MOSSCO suite. Based on the parameterization by Breugem and Holthuijsen 2007, significant wave height and peak wave period are estimated in terms of local water depth, wind speed and fetch length. This wave data enable the inclusion of wave effects especially for idealized 1D water column studies, e.g. the consideration of erosion processes due to wave-induced bottom stresses. Coupling to 3D ocean models and the calculation of additional wave-induced momentum forces there, following either the Radiation stress or Vortex Force formulation (Moghimi et al., 2013), is possible as well. For the inclusion of wave–wave or wave–current interaction in realistic 3D applications, the coupling to a more advanced third generation wind wave model like SWAN, WaveWatch III or Wave Atmospheric Model (WAM) would be necessary.

### 3.2 Input/Output utilities

The input and output (I/O) utilities include general purpose coupling functionalities that deal with boundary conditions, provide a restart facility, add surface, lateral and point source fluxes (lower left branch in Fig. 4).

#### 3.2.1 NetCDF output

This component of MOSSCO provides an output facility `netcdf_component` for any data that is communicated in the coupling framework. The component writes one- to three-dimensional time sliced data into a NetCDF (Rew and Davis, 1990) file and adds metadata on the simulation to this output. Multiple instances of this component can be used within a simulation, such that output of different variables, differently processed data, and output at various output time steps can be recorded. The output component is fully parallelized with a grid decomposition inherited from one of the coupled science or data components.

In order to reduce interprocess communication during runtime, each write process considers only the part of the data (its data tile) that resides within its compute domain. This comes at a cost to the user, who has to postprocess the output tiles to combine for later analysis; a python script is provided with MOSSCO that takes care of joining tiled files.

The output component also adds metadata that is collected from the system and the user environment at the creation time of the output files. Diagnostics about the processing element and run time between output steps are recorded. The structure of the

15 NetCDF output follows the Climate and Forecast (CF, Eaton et al., 2011) convention for physical variables, geolocation, units, dimensions and methods modifying variables. When (mostly biological) terms are not available in the controlled vocabulary of CF, names are built to resemble those contained in the standard.

#### 3.2.2 NetCDF input

The `netcdf_input_component` of MOSSCO reads from NetCDF files and provides the file content wrapped in ESMF

data structures (fields) to the coupling framework. It inherits its decomposition from other components in the coupled system. Data can be read from a single file for the entire domain or from distributed files for all decomposed compute elements separately. Upon reading of data, fields can be renamed and filtered before they are passed on to the coupled system.

The input component is typically used to initialize other components, for restarting, to provide boundary conditions, and for assimilating data into the coupled system. The input facility supports interpolation of data in time upon reading the data,

with nearest, most recent, and linear interpolation. It also supports reading climatological data and translates the climatological timestamp to a simulation present time stamp in the coupling framework.

### 3.3 MOSSCO connectors and mediators

Information in the form of ESMF states that contain the output fields of every component are communicated to the ESMF driver; requests for data by every component are also communicated to the ESMF driver component. MOSSCO connectors

are separate components that link output and requested fields between pairwise coupled components. MOSSCO informally

distinguishes between connector components that do not manipulate the field data on transfer at all (or only slightly), and mediator components that extract and compute new data out of the input data.

### 3.3.1 Link, copy and nudge connectors

The simplest and default connecting action between components is to share a reference (i.e. a link) to a single field that resides in memory and can be manipulated by each component; in contrast, the `copy_connector` duplicates a field at coupling time. The consideration of a link or copy connector is important for managing the data flow sequence in a coupled system: the copy mechanism ensures that two coupled components work on the same lagged state of data, whereas the link mechanism ensures that each component works on the most recent data available.

The `nudge_connector` is used to consolidate output from two components by weighted averaging of the connected fields. It is typically used as a simple assimilation tool to drive model states towards observed states, or to impose boundary conditions.

These connectors can only be applied between components that run on the same grid (but maybe with a different subdomain decomposition). The `link_connector` can only be applied between components with an identical subdomain decomposition so that the components have access to the same memory. Components on different grids require regridding, which is currently under development in MOSSCO.

### 3.3.2 Transport connector

A model component qualifies as a transport component when it offers to transport an arbitrary number of tracers in its numerical grid; this facility is present, for example, in the current `gotm_component` and `getm_component`. The `transport_connector` collects state variables to be transported from any coupled component and communicates this collection to the hydrodynamic component based on the availability of both, the tracer concentrations as well as their rate of vertical movement independent of the water currents. This connector is usually called only once per coupled pair of components during the initialization phase.

### 3.3.3 Mediators for soil–pelagic coupling

One aspect of the generalized coupling infrastructure in MOSSCO is the use of connecting components that mediate between technically or scientifically incompatible data field collections. The soil–pelagic coupling of biogeochemical model components with a variety of different state variables raises the need for these mediators. The use of mediators leaves the level of data aggregation and dis-aggregation, and unit conversion to the coupling routine, instead of requiring specific output from a model component depending on its coupling partner component.

For soil–pelagic (or benthic–pelagic) coupling, the `soil_pelagic_connector` mediates the soil biogeochemistry output towards the pelagic ecosystem input and the `pelagic_soil_connector` mediates the pelagic ecosystem output towards the soil biogeochemistry input, e.g.: (i) dis-aggregation of dissolved inorganic nitrogen to dissolved ammonium and dissolved nitrate (ii) filling missing pelagic state fields for phosphate using the Redfield-equivalent for dissolved inorganic

nitrogen (iii) calculation of the vertical flux of particulate organic matter (POM) from the water column into the sediment depending on POM concentrations in the near-bottom water, its sinking velocity and a sedimentation efficiency depending on the near-bottom turbulence. The effective vertical flux is communicated into the pelagic ecosystem component to budget the respective loss, and is communicated to the soil biogeochemistry component to account for the respective new mass of POM. The mediator also handles (iv) dis-aggregation of a single oxygen concentration (allowing positive and negative values) into dissolved oxygen concentration, if positive, and dissolved reduced substances, if negative. (v) aggregation of pelagic POM composition (variable nitrogen to carbon ratio) into fixed stoichiometry POM pools in the soil biogeochemistry.

## 4 Selected applications as feasibility tests and use cases

MOSSCO was designed for enhancing flexibility and equitability in environmental data and model coupling. These design goals have been helpful in generating new integrated models for coastal research with applications at different marine stations (1D), transects (2D), and sea domains (3D). Below, we describe from a user perspective the added value and success of the design goals obtained from using MOSSCO in selected applications; here, the focus is not on the scientific outcome of the application (these are described elsewhere by, e.g., Nasermoaddeli et al. 2017, Slavik et al. (subm.), and Kerimoglu et al. 2017). All setups described in the use cases are available as open source (with limited forcing data due to space and bandwidth constraints).

### 4.1 Helgoland station

The seasonal dynamics of nutrients and turbidity emerges from the interaction of physical, ecological and biogeochemical processes in the water column and the underlying sea floor. We resolve these dynamics in a coupled application for a 1D vertical water column for a station near the German offshore island Helgoland. Average water depth around the island is 25 m; tidal currents are affected by the M2 and S2 tides with a characteristic spring–neap cycle, with current velocity not exceeding 1 $\mathrm{m\,s^{-1}}$.

The Helgoland 1D application is realized by a coupled system consisting of GOTM hydrodynamics, the pelagic FABM component with a nutrient–phytoplankton–zooplankton–detritus (NPZD) ecosystem model (Burchard et al., 2005) and two SPM size classes, interacting with the erosion and sedimentation module, the sediment component with the OMEXDIA_P early diagenesis submodel, and coupler components for soil–pelagic, pelagic–soil and tracer transport. This system and setup is described in more detail by Hofmeister et al. (2014).

Simulations with this application show a prevailing seasonal cycle in the model states (Fig. 5). Dissolved nutrients (referred as dissolved inorganic nitrogen) are taken up by phytoplankton, which fills the pool of particulate organic nitrogen during the spring bloom (Fig. 5d). The particulate organic matter sinks into the sediments, where it is remineralized along axis, sub-oxic and anoxic pathways; denitrification, for example, shows a peak in late summer (Fig. 5b). At the end of a year, nutrient concentrations are high in the sediment and diffuse back into the water column up to winter values of 20–25 $\mathrm{mmol\,m^{-3}}$. The seasonal variation of turbidity is a result of higher erosion in winter and reduced vertical transport in summer (Fig. 5c).

## 4.2 Idealized coastal 2D transect

The coastal nitrogen cycle is resolved in an idealized coupled system for a tidal shallow sea. This two-dimensional setup represents a vertically resolved cross-shore transect of 60 km length and 5–20 m water depth and has been used by Hofmeister et al. (2017) to simulate sustained horizontal nutrient gradients by particulate matter transport towards the coast. Within the MOSSCO coupling framework, the 2D transect scenario additionally provides insights into horizontal variability of erosion/sedimentation and benthic biogeochemistry. Its coupling configuration builds on the one used for the 1D station Helgoland

(Section 4.1); the water-column hydrodynamic model GOTM, however, is replaced by the 3D model GETM; a local wave component and data components for open boundaries and restart has been added.

Fig. 6 shows exchange fluxes between the water column and the sediment for one year of simulation. The simulation of turbidity, as a result of pelagic SPM transport and resuspension by currents and wave stress, provides the light climate for the pelagic ecosystem. The flux of particulate organic carbon (POC) into the sediment reflects bloom events in summer during

calm weather conditions. Macrobenthic activity in the sea floor brings fresh organic matter into the deeper sub-oxic layers of the sediment, where denitrification removes nitrogen from the pool of dissolved nutrients. The coupled simulation reveals decoupled signals of benthic respiration, denitrification and nutrient reflux into the water column, which is not resolved in monolithically coded regional ecosystem models of the North Sea (Lorkowski et al., 2012; Daewel and Schrum, 2013).

## 4.3 Southern North Sea bivalve ecosystem applications

A Southern North Sea (SNS) domain was used in two studies concerning the effects of bivalves on the pelagic ecosystem. Slavik et al. (subm.) investigated how the accumulation of epifauna on wind turbine structures (Fig. 7d) impacts pelagic primary production and ecosystem functioning in the SNS at larger spatial scales. This study is the first of its kind that extrapolates ecosystem impacts of anthropogenic offshore wind farm structures from a local to a regional sea scale. The authors use a MOSSCO coupled system consisting of the hydrodynamic model GETM, the ecosystem model MAECS as described by

Kerimoglu et al. (2017), the transport connector, the filter feeder component, and several input and output components (Fig. 7e). They assess the impact of anthropogenically enhanced filtration from blue mussel (*Mytilus edulis*) settlement on offshore wind farms that are planned to meet the 40-fold increase in offshore wind electricity in the European Union until 2030; they find a small but non-negligible large-scale effect in both phytoplankton stock and primary production, which possibly contributes to better water clarity (Fig. 7f).

Biological activities of macrofauna on the sea floor mediate suspended sediment dynamics, at least locally. In the study by Nasermoaddeli et al. (2017), the large-scale biological contribution of benthic macrofauna, represented by the key species *Fabulina fabula* (Fig. 7a), to suspension of sediment was investigated. Simulation results for a typical winter month revealed that SPM is increased not only locally but beyond the mussel inhabited zones. This effect is not limited to the near-bed water layers but can be observed throughout the entire water column, especially during storm events (Fig. 7c). In this coupled

application, the hydrodynamic model GETM, the pelagic ecosystem component with three SPM size classes, the erosion–

sedimentation and benthic mediation components, several input and one output components, and the transport connector were used (Fig. 7d).

## 4.4 Exemplary workflow

For the SPM bivalve example above (Nasermoaddeli et al. 2017 and Fig. 7c), the coupled system contains 13 modular components: the hydrodynamic `getm_component` and `simplewave_component`, a pelagic `fabm_component`, the benthic erosion/sedimentation `erosed_component` and `benthos_component`, four input and one output component, and the

default `link_connector`, the `nudge_connector` as well as the `transport_connector`. Each of these 13 components is involved in at least one pairwise coupling described in a `couplings` section of the YAML coupling configuration (Fig. 3b). This coupled application is to be deployed in sequential mode on the same set of compute resources for all 13 components.

The horizontal spatial representation and domain decomposition are provided by the grid that is created in the hydrody-

namic model and that is communicated to the wave, pelagic ecosystem, benthic and input components; this is achieved by specifying the hydrodynamic model as a dependency of these components (`dependencies` in Fig. 3b) Four instances of the `netcdf_input_component` (see 3.2.2 and `instances` in Fig. 3b) are created to provide macrofauna forcing, lateral open ocean boundaries, rivers fluxes, and restart information from netCDF files. In the first of two initialization phases, the output components and the hydrodynamic component are initialized first, as they have not dependencies. Dependent compo-

nents then receive the spatial grid information from the hydrodynamic component. All components advertise what information they can provide (e.g., a certain quantity) and what information they need (e.g., grid information) in the coupled system.

In the second initialization phase, the `transport_connector` ensures that all fields from the ecosystem component are made available in the hydrodynamic component for advection and diffusion. For all other pairwise couplings, the `link_connector` communicates advertised data from a sending component as a pointer to the receiving component; passing pointers to data in-

stead of copies of the data itself is only possible in sequential mode and on identical grids . In the restart phase, additional initialization data is communicated to all components implementing this (optional) phase; here, the bed mass and SPM concentrations are updated in the ecosystem and erosion components via a coupling to an instance of the input component that reads data from disk that was created in prior model runs ("restart").

In the run phase, all pairwise couplings are called in the same order as during the initialization phase. First, the connector

(or coupler) is called to synchronize the two components' data, then each of the coupled components in this pairwise coupling is executed for the minimum time interval to the next coupling time step of the involved components (see Fig. 2). With the boundary conditions read with the input component from file at each coupling interval, the SPM fields that reside in the ecosystem component are updated by way of connecting these components with the `nudge_connector`. Finally, at the end of a simulation, all output components are run once more to ensure that the final state of the system is recorded; then, all

components go through their finalize phase and clean up reserved memory.

## 5 Discussion and Outlook

In merging existing frameworks that address distinct types of modularity and by developing a super-structure for making the multi-level coupling approach applicable in coastal research, the MOSSCO system largely meets the design goals *flexibility* and *equitability*. In doing so, structural deficiencies of legacy models and the need for practical compromises became very apparent.

For legacy reasons, *equitability* is the harder to achieve design goal. Both the distribution of compute resources as well as the spatial grid definition can be in principle determined by any one of the participating components; de facto, in marine or aquatic research, they are prescribed by the hydrodynamic models that have so far not been enabled to inherit a grid specification or a resource distribution from a coupler or coupled system. With the ongoing development and diversification of hydrodynamic models, and no immediate benefit for the different physical models to outsource grid/resource allocation, this situation is not likely to change. MOSSCO compromises here by its flexible grid inheritance scheme and with the grid provisioning component that delivers this information to the coupled system whenever a hydrodynamic component is not used.

Beyond grid/resource allocation, however, the *equitability* concept is successfully driving independent developments of submodules. We found that experts in one particular model, e.g. the erosion module, could rely on the functionality of the other parts of the system without having to be an expert themselves in all of the constituent models in the coupled application. The limitations to this black-box approach became evident in the scientific application and evaluation of the coupled model system, which was only possible when a collaboration with experts in these other model systems was sought. By taking away the inaccessibility barrier and by enforcing clear separation of tasks the modular system stimulated a successful collaboration. Sustained granularity also helped to align with ongoing development in external packages. These can be integrated fast into the coupled system, which does not rely on specific versions of the externally provided software unless structural changes occur. Long-term supported interfaces on the external model side facilitate MOSSCO being up-to-date with e.g. the fast evolving GETM and FABM code bases.

When legacy codes were equipped with a framework-agnostic interface we encountered four major difficulties:

1. For organizing the data flow between the components, MOSSCO uses standard names and units compatible with the infrastructure and library of standard names and units provided in the pelagic component for the FABM framework (mostly modelled on CF). Other components, such as the BMIs of wrapped legacy models, do not provide such a standard name in their own implementation, and in particular, often do not adhere to a naming standard. We found ambiguity arising, e.g., with temperature to be represented as `temperature` vs. `sea_water_temperature` vs. `temperature_in_water`. While this can be resolved based on CF for temperature, most ecological and biogeochemical quantities currently lack a consistent naming scheme. The forthcoming GSN ontology (building on CSDMS names, Peckham, 2014) could adequately address this coupling challenge.

2. Deep subroutine hierarchies of existing models made it difficult to isolate desired functionality from the structural external overhead. In one example, where a single functional module was taken out of the context of an existing third-party coupled system, the module depended on many routines dispersed throughout that third-party system repository.

3. Components based on standalone models are developed and tested with their own I/O infrastructure and typically supply a BMI implementation only for part of their state and input data fields. A new, coupled application or data provisioning/request within a coupled system can therefore easily require a change in a model's BMI. The implementation potential input and output for all quantities, including replacement of the entire model-specific I/O in the BMI is therefore desirable for new developments and refactoring.

4. Mass and energy need to be conserved across the coupled components. Mediators communicate conservatively regridded mass and energy fluxes into pairs of coupled components. These fluxes then need to be appropriately integrated by the coupled components, even when their internal time discretization differs and for asynchronous scheduling that can incur different coupling timesteps. Conservative integration of exchanged mass and energy fluxes cannot automatically be ensured by the coupling system, and the user has to carefully consider timesteps in the preparation of the coupling setup.

Efforts in making legacy models coupleable, either for MOSSCO or similar frameworks, however, can have additional benefits besides the immediate applicability in an integrated context. coupleability strictly demands for communication of sufficient metadata, which stimulates the quality and quantity of documentation and of scientific and technical reproducibility of legacy models. Indeed, transparency has been greatly increased by wrapping legacy models in the MOSSO context. All participating components performed introspection and leveraging of a collection of metadata at assembly time of the coupled application and during output. Transparency is expected to be continuously increasing by new coupling demands and more generous metadata provisioning from wrapped science models. MOSSCO is moving towards adopting the Common Information Model (CIM) that is also required by Climate Model Intercomparison Project (CMIP) participating coupled models (Eyring et al., 2016).

With a current small development base of twelve contributors, the openness concept of MOSSCO in terms of including contributions from outside the core developer team has not yet been tested; in the categorization by de Laat (2007) internal governance with simple structure is sufficient at this size. Formally, external contributions can be included in MOSSCO by way of contributor license agreements. The openness concept has been useful in instigating discussions about the need for explicit (and preferably open) licensing of related scientific software and data as demanded in current open science strategies (e.g., Scheliga et al., 2016).

Scalability in MOSSCO applications depends on the scalability of the coupled model components and on the potential overhead of the coupling infrastructure. Strong scaling experiments were performed with a coupled application using GETM, FABM with MAECS ($\approx$20 additional transported 3D tracers), and FABM with OMEXDIA_P—including bidirectional benthic–pelagic coupling—on Jureca (Krause and Thörnig, 2016). They show linear (perfect) scaling from 100 to 1000 cores, and a small leveling-off (to 85% of perfect scaling) at 3000 cores. We have not observed loss of compute time due to the infrastructure and superstructure overhead of ESMF, which remained below 0.1% in the run phase of the scaling experiment. A typical operational computation speed achieved, e.g., in the bivalve windfarm application (Sect. 4.3, 175000 grid cells) on 192 processors is 2000 computed hours per elapsed wall clock hour: such a performance allows decadal up to multidecadal simulations. One of the identified bottlenecks (that varies strongly with the HPC system used) is data transfer from memory

to disk: this will be in the future addressed by the use of parallel NetCDF and/or leveraging the XML I/O Server (XIOS, Meursedoif, 2013).

Multi-component systems may also suffer from low acceptance by the research community. They are much harder to be implemented and maintained by individual groups, where researchers solve coastal ocean problems of a large range of complexity, from purely hydrodynamic applications via coupled hydrodynamic–sediment dynamic applications to fully coupled systems. Many academic problems focus on specific mechanisms and thus do not require the complete and fully coupled modular system, such that the application of the full system might mean a large structural overhead and additional workload. There is, however, the necessity of following a holistic approach when tackling grand research questions in environmental science such as related to system responses to anthropogenic intervention. Yet, it is not clear whether the bottom-up approach of many interacting modular components leads to an emergent system behaviour that is desirable and exhibits new insights or whether the system gets tangled up in coupling complexity.

As evident form the test cases (Sect. 4), MOSSCO also encourages coupled applications that are far from a complete system level description. With few coupled components, the technical threshold to getting an application running on an arbitrary system is relatively low. The user can reach a fast first success. MOSSCO provides a full documentation, step by step recipes, and a public bug tracker; it adopts abundant error reporting from ESMF and a fail fast design that stops a coupled applications as soon as a technical error is detected (Shore, 2004). Usability is especially high due to an available master script that compiles, deploys, and schedules a coupled application. To address a wide range of users, the system is designed to run on a single processor or on a user's laptop equally well as on a high-performance computer using several thousand compute nodes.

An obvious advantage of modular coupling is the opportunity to bridge the gap between different scientific disciplines. It allows in principle to combine, e.g., hydrodynamic models from oceanography with sediment transport models from coastal engineering. Thus different experts can work on their individual models but benefit from all others' progress. This seeming advantage, however, poses also a drawback for modular coupling approaches. An initial effort which is necessary for individual models to meet the requirements of a modular modelling framework has to be invested. This will only happen if there is either an urgent pressure to include specific model capabilities, which will be difficult to include otherwise, or if convincing examples of possible benefits can be presented. It cannot be expected that the coastal ocean modelling community will agree about one coupler or one way of interfacing modules, such that it will still require considerable implementation work to transfer a module from one modular system to another. To solve this problem, coupling standards need to become more general, but in turn this might even increase the structural overhead in using these systems.

For certain applications it might be preferable for different reasons to hardwire submodels and exercise strong control over such a monolithic coupled system. But at the least, such submodels should be made coupleable by following the minimal requirements set forth by the BMI specification. This ensures that the monolithic model system or parts of it can be reused or expanded in a more modular way. And by strictly separating the BMI from any framework specific CMI specification, the effort spent on wrapping an existing model, or on equipping a new model with a basic model interface is not tied to a particular coupling framework, or even a particular coupling framework technology. A model that follows BMI principles will be more

easily interfaced to other models no matter what coupler is used. Wrapped legacy models from MOSSCO can thus be useful in non-ESMF contexts, as well; and models with an existing BMI can be integrated in MOSSCO more easily, in turn.

One demand for integrative modelling, which is likely best practised in open and flexible system approaches, arises from current European Union legislation. The Water Framework Directive and the Marine Strategic Planning Directive require the description of marine environmental conditions and the development of action plans to achieve a good environmental status. These objectives can initially be met by a monitoring program to determine present-day conditions but ultimately rely on numerical model studies to evaluate anthropogenic measures. This ecosystem based approach to management (e.g., Ruckelshaus et al., 2008) demands modelling systems which are capable of taking into account hydrodynamics, biogeochemistry, sedimentology and their interactions to properly describe the environmental status. As further legal requirements can be expected for many coastal seas worldwide, numerical modelling systems applied for this task need to be flexible in terms of integrating additional (e.g., site-specific) processes. In this ongoing process, the initial effort of creating a modular system may be the only way forward that can take into account all relevant processes in the long run.

## 5.1 Outlook

The suite of components provided or encapsulated so far meets the demands that were initially formulated by our users; they already allow for a wide range of novel coupled applications to investigate the coastal sea. To stimulate more collaboration, however, and to bring forward a general "ecosystem" of modular science components, several legacy models could interface to MOSSCO components in the near future by building on complementary work at other institutions. For example, the Regional Earth System Model (RegESM, Turuncoglu et al., 2013; Turuncoglu and Sannino, 2017) provides ESMF interfaces for MITgcm, ROMS and WAM, amongst others. Convergence of the development of MOSSCO and RegESM is feasible in the near term. Also, the recently developed Icosahedral Non-Hydrostatic Atmospheric Model (ICON Zängl et al., 2015) is currently being equipped with an ESMF component model interface.

Once ESMF interfaces have been developed for a legacy model, it is desirable that these developments move out of the coupler system and are integrated into the development of the legacy model itself. This has been successfully achieved with the ESMF interface for the hydrodynamic model GETM, which is now distributed with the GETM code. Much of the utility layer developed in MOSSCO, or likewise in MAPL or in the ESMF extension of the WRF model, are expected to be propagated upstream into the framework ESMF itself.

The interoperability of current coupling standards will increase. While currently there are three flavors of ESMF: basic ESMF as in MOSSCO, ESMF/MAPL as in the GEOS-5 system, or ESMF/NUOPC as in the RegESM, only a minor effort would be required to provide the basic ESMF and ESMF/MAPL implementations with a NUOPC cap and make them interoperable with the entire ESMF ecosystem. Even a coupling of ESMF based systems to OASIS-MCT based systems has been proposed; and investigation is ongoing on a coupling of MOSSCO to the formal BMI for CSDMS.

# 6    Conclusions

We problematized both the primacy of hydrodynamic models and the limited modularity in coupled coastal modelling that can stand in the way of developing and applying novel and diverse biogeochemical process descriptions. Such developmental potential is likely needed to progress towards holistic regional coastal systems models. We presented the novel Modular System for Shelves and Coasts (MOSSCO), that is built on coupling concepts centered around equitability and flexibility to resolve the issue of obstructed modularity. These concepts bring about also openness, usability, transparency and scalability. MOSSCO as an actual Fortran implementation of this concept includes the wrapped Framework for Aquatic Biogeochemical Models (FABM) and a usability layer for the Earth System Modeling Framework (ESMF).

MOSSCO's design principles emphasize basic coupleability and rich meta information. Basic coupleability requires that models communicate about flow control, compute resources, and about exchanged data and metadata. We demonstrated that the design principles flexibility and equitability enable the building of complex coupled models that adequately represent the complexity found in environmental modelling. In this first version, the MOSSCO software wrapped several existing legacy models with basic model interfaces (BMI); we added ESMF-specific component model interfaces (CMI) to these wrappers and other models and frameworks to build a suite of ESMF components that when coupled represent a small part of a holistic coastal system. These components deal with hydrodynamics, waves, pelagic and sediment ecology and biogeochemistry, river loads, erosion, resuspension, biotic sediment modification and filter feeding.

In selected applications, each with a different research question, the applicability of the coupled system was successfully tested. MOSSCO facilitates the development of new coupled applications for studying coastal processes that extend from the atmosphere through the water column into the sea bed, and that range from laboratory studies to 3D simulation studies of a regional sea. This system meets an infrastructural need that is defined by experimenters and process modellers who develop biogeochemical, physical, sedimentological or ecological models at the lab scale first and who would like to seamlessly embed these models into the complex coupled three-dimensional coastal system. This upscaling procedure may ultimately support also the global Earth System community.

*Code and data availability.* The MOSSCO software is licensed under the GNU General Public License 3.0, a copyleft open source license that allows the use, distribution and modification of the software under the same terms. All documentation for MOSSCO is licensed under the Creative Commons Attribution Share-Alike 4.0 (CC-by-SA), a copyleft open document license that allows use, distribution and modification of the documentation under the same terms.

Development code and documentation are currently primarily hosted on Sourceforge (https://sf.net/p/mossco/code) and mirrored on Github (https://github.com/platipodium/mossco-code). The release version 1.0.1 is permanently archived on Zenodo and accessible under the digital object identifier doi:10.5281/zenodo.438922. All wrapped legacy models are open source and freely available from the developing institutions; free registration is required for accessing the Delft3D system at Deltares. Selected test cases are available from a separate Sourceforge repository https://sf.net/p/mossco/setups, where all of the data on which the presented use cases are based are freely available,

with the exception of the meteorological forcing fields. These are, for example, available at request online at http://www.coastdat.de, from the coastDat model based data base developed for the assessment of long-term changes by Helmholtz-Zentrum Geesthacht (Geyer, 2014).

*Author contributions.* C.L., R.H., K.K., H.N developed the MOSSCO components (CMI) and wrappers (BMI). K.W, C.L., and K.K. designed the coupling philosophy, C.L. developed the user interface and the utility library. K.W., H.N., R.H., O.K. and C.L. carried out and analysed simulations, based on contributions from all authors. C.L., K.W. and R.H. wrote the manuscript with contributions from all other authors.

*Competing interests.* The authors declare that they have no conflict of interest.

*Acknowledgements.* MOSSCO is a project funded under the Küstenforschung Nordsee–Ostsee programme of the Forschung für Nachhaltigkeit (FONA) agenda of the German Ministry of Education and Science (BMBF) under grant agreements 03F0667A, 03F0667B, and 03FO668A. This research contributes to the PACES II programme of the Hermann von Helmholtz-Gemeinschaft Deutscher Forschungszentren. Further financial support for K.K. and H.B. was provided by the Collaborative Research Centre TRR181 on Energy Transfers in Atmosphere and Ocean funded by the German Research Foundation (DFG). The authors gratefully acknowledge the computing time granted by the John von Neumann Institute for Computing (NIC) and provided on the supercomputer JURECA at at Forschungszentrum Jülich. We thank those MOSSCO developers that are not co-authors of this paper, amongst them Markus Kreus, Ulrich Körner and Niels Weiher, and acknowledge the support of Wenyan Zhang and Kaela Slavik in preparing the model setups. This research is based on tremendous efforts by the open source community, including but not limited to the developers of Delft3D, GETM, GOTM, FABM, ESMF, OpenMPI, Python, GCC and NetCDF who share their codes openly.

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

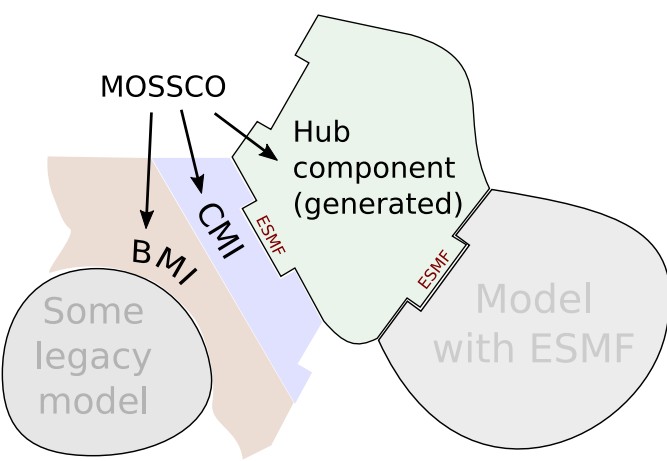

**Figure 1.** MOSSCO's adoption of legacy code follows the two-layer paradigm of BMI/CMI (basic model interface/ component model interface) suggested by Peckham et al. (2013). An existing legacy code (illustrated by "some model") is enhanced by model-specific code that exhibits basic coupling functionality ("BMI") and is framework agnostic. In a second step, a component ("CMI") is added, that uses the BMI interface in the specific application programming interface of the coupling framework. In addition to model interfaces that can be used in MOSSCO-independent contexts, MOSSCO provides coupling concepts and working implementations for coupled applications.

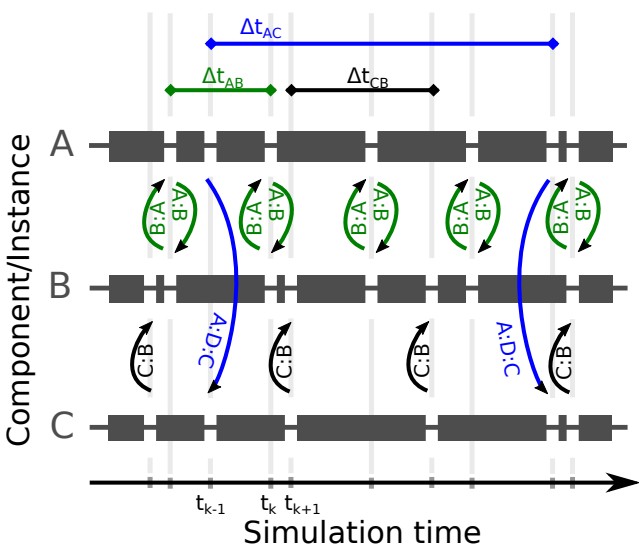

**Figure 2.** Scheduling of three coupled component instances A, B, C and their data exchanges according to a pairwise coupling specification (see Fig. 3b), shown along a simulation time axis, which is independent of the type of (sequential or concurrent) deployment. Note how each individual component instance has varying run lengths resulting from the interference of all coupling intervals with this component. The time steps of the (anonymous) scheduler component $t_{k+n}$ (grey bars) vary according to the interference pattern of all coupling intervals. Coupling specification (Fig. 3): A couples bidirectionally to B at interval $\Delta t_{AB}$ (green), A couples unidirectional to C via a coupler D at interval $\Delta t_{AC}$ (blue), C couples unidirectionally to B at interval $\Delta t_{CB}$ (black).

```
# This YAML text file specifies
# a minimal coupling for MOSSCO
# using only two components
# and default couplers/intervals

coupling:                    (a)
    components:
        - A
        - B
```

```
# This YAML text file specifies
# a more elaborate coupling for MOSSCO
# using three components and two couplers

dependencies:
    - B: A     # run B after A      (b)
    - C: B     # run C after B

instances:
    - C: A     # run C as instance of A

coupling:
    - components:
        - A              # send component
        - B              # receive component
      interval: 10 h  # Dt_AB in Fig. 2
    - components:
        - B
        - A
      interval: 10 h
    - components:
        - C
        - B
      interval: 14 h  # Dt_CB in Fig. 2
    - components:
        - A
        - D    # this is the coupler
        - C
      interval: 34 h  # Dt_AC in Fig. 2
```

```bash
#! /bin/bash

1  export MOSSCO_DIR=$HOME/MOSSCO/code           (c)
2  export MOSSCO_SETUPDIR=$HOME/MOSSCO/setups
3  export NETCDF=NETCDF4

4  git clone --depth=1 git://git.code.sf.net/p/mossco/code $MOSSCO_DIR
5  git clone --depth=1 git://git.code.sf.net/p/mossco/setups $MOSSCO_SETUPDIR

6  make -C $MOSSCO_DIR external   # download external codes

7  mkdir -p $HOME/opt/bin
8  export PATH=$PATH:$HOME/opt/bin
9  ln -sf $MOSSCO_DIR/scripts/mossco.sh $HOME/opt/bin/mossco # "installation"

10 cd $MOSSCO_SETUPDIR/helgoland  # choose a Helgoland 1D setup
11 mossco jfs                     # starts a 1D pelagic-sediment simulation
```

**Figure 3.** Examples of coupling configurations (a,b), and the steps from installation to deployment (c). The configurations exhibit a minimal default coupling specification (a) and a more complex one (b, cf. Fig. 2) that makes use of dependencies, instantiation and different coupling intervals. The line numbered installation steps (c) include environment variable specification (export), download of the system with git, loading additional external models with make, installation of the mossco executable script and finally deployment of the coupling specification jfs in a predefined setup called "helgoland", a 1D station near the island Helgoland in the Southern North Sea.

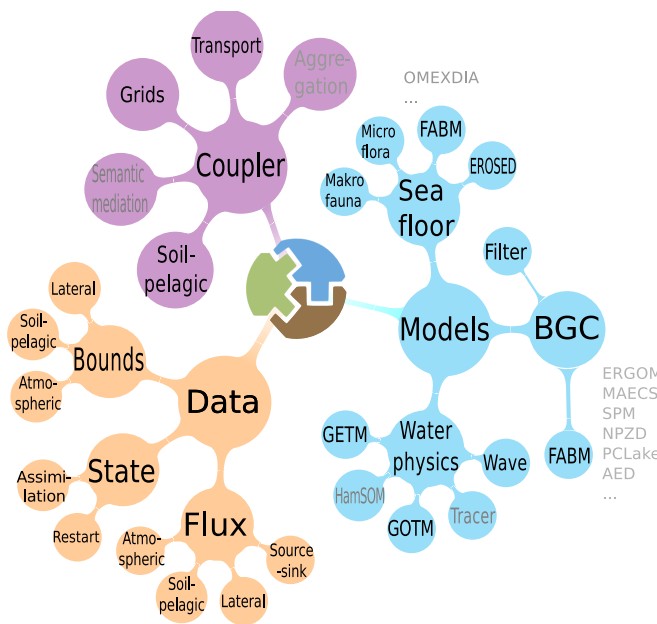

**Figure 4.** Modular components of MOSSCO. The blue branch collects newly created submodels and components that wrap around legacy codes; the violet branch collects coupling functionalities and the orange branch the Input/Output utilities.

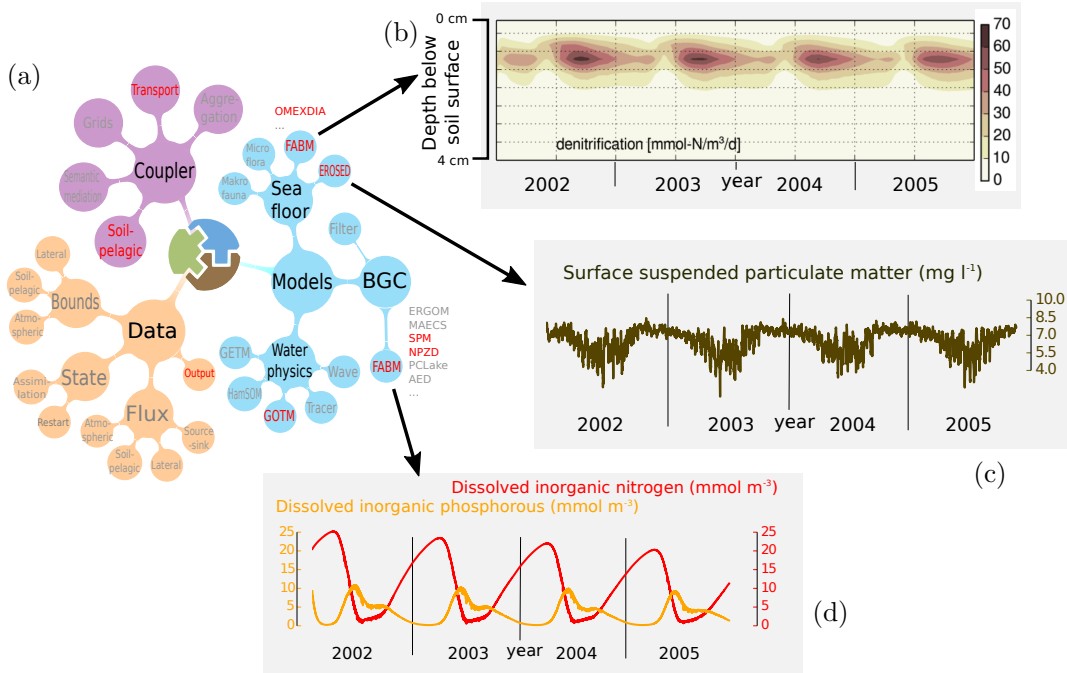

**Figure 5.** Coupling setup and exemplary results from a 1D system simulating the nutrient and SPM dynamics near the island of Helgoland, Germany with soil–pelagic coupling from 2002 to 2005. (a) Coupling setup with seven ESMF components (highlighted in red, leaves) and three FABM submodels (side text); (b) soil denitrification rate; (c) surface SPM dynamics resulting from EROSED and pelagic FABM/SPM; (d) middle water column nitrogen and phosphorous dynamics from pelagic FABM/NPZD.

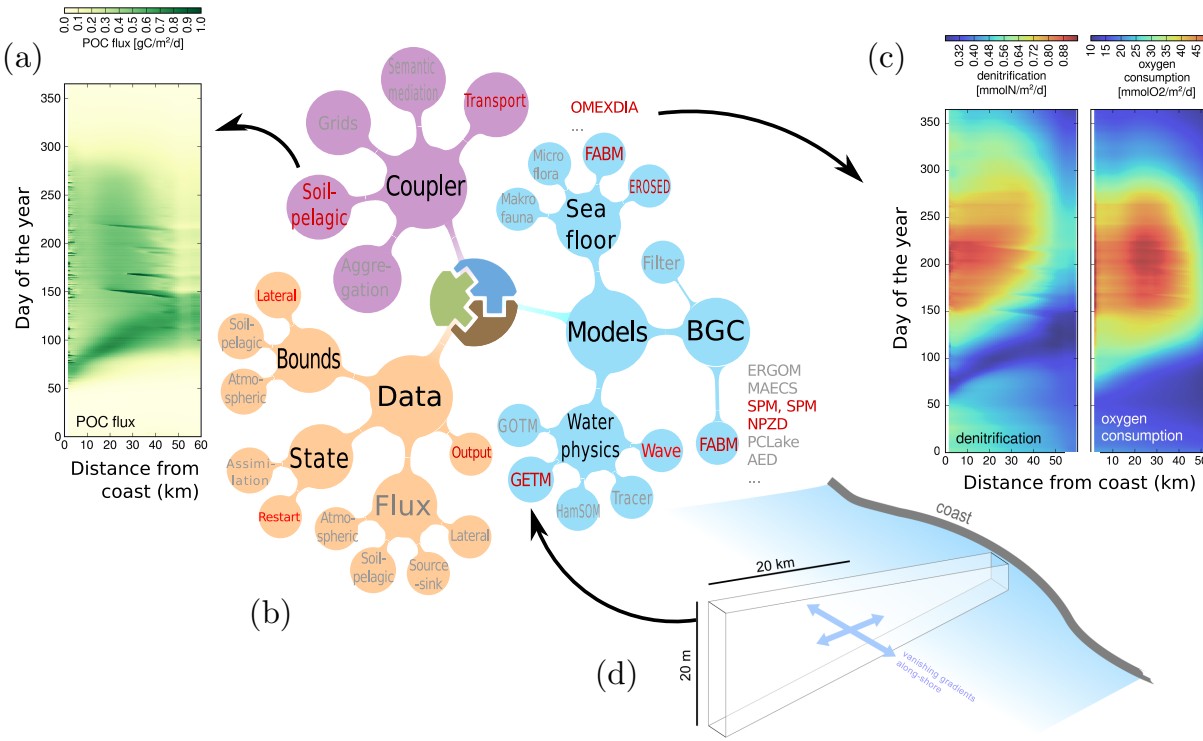

**Figure 6.** A 2D idealized cross-shore transect off the German coastline is used to investigate the feedback loop between estuarine circulation, sediment transport and nutrient cycling across the benthic–pelagic interface. (a,c) Hovmöller diagrams showing the soil–pelagic fluxes of particulate organic carbon (POC) and the soil BGC denitrification and oxygen consumption rates for the 60 km long transect. (b) Coupling diagram including components for hydrodynamics, erosion/sedimentation, waves, pelagic ecology and suspended particles, and soil ecology. This example uses both ESMF modularity (the components) as well as FABM modularity (the different ecological/biogeochemical models within the pelagic and sediment environmental components). (d) spatial setup of the idealized 2D cross-shore transect.

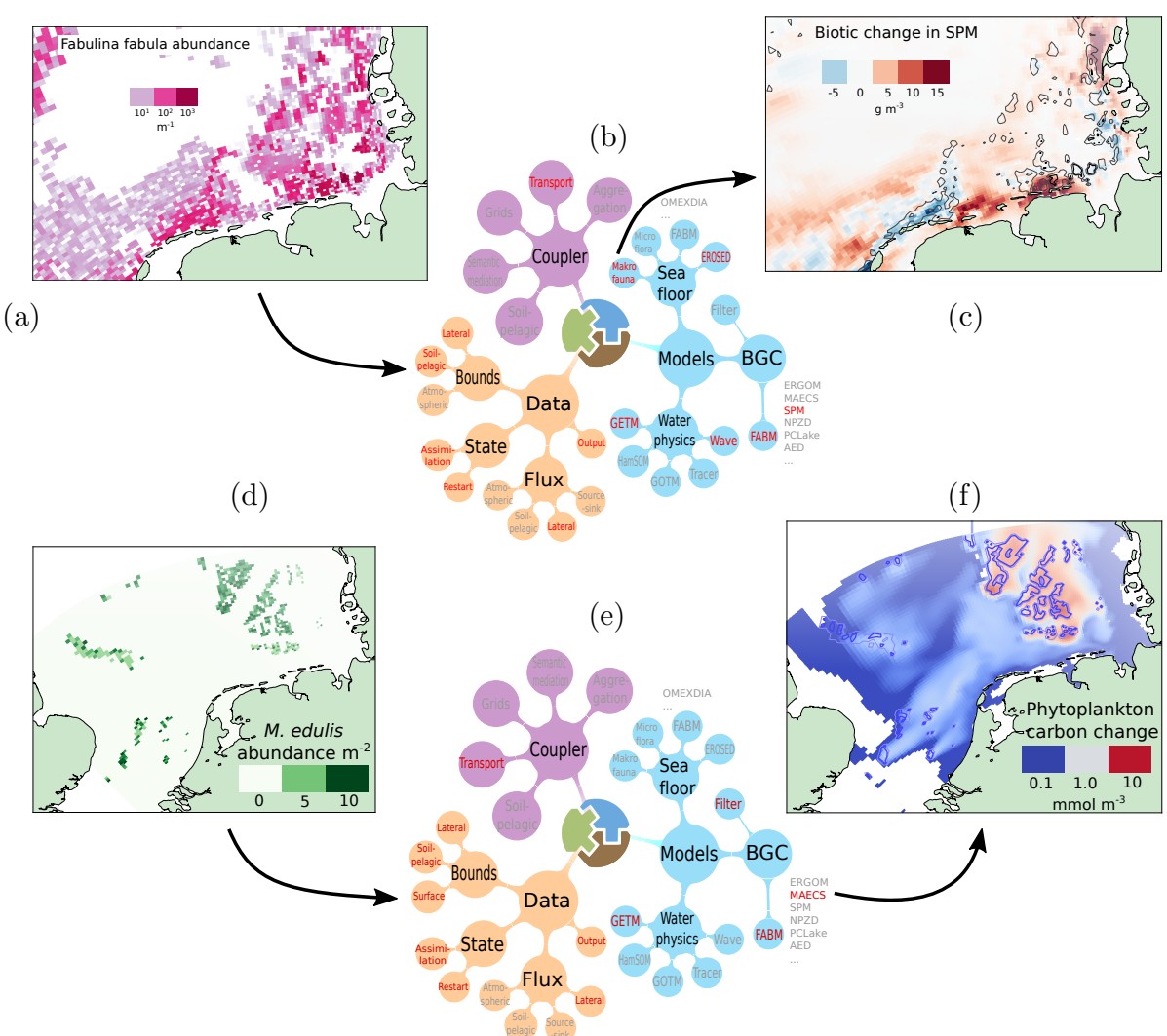

**Figure 7.** Building flexible applications with MOSSCO. Two bivalve related scientific applications are showcased: Nasermoaddeli et al. (2017) investigated the effect of bottom-dwelling *Fabulina fabula* (a, showing parts of the southern North Sea) on suspended sediment concentration (c) with a coupled application integrating hydrodynamics, three pelagic SPM classes in the ecosystem model, mediation of erodibility by benthic bivalves and an explicit description of bed erosion and sedimentation (b), cf. Section 4.4 and Fig. 5. Slavik et al. (subm.) investigated the effect of epistructural *Mytilus edulis* (d) on phytoplankton concentration (f) with a coupled application integrating hydrodynamics, the FABM/MAECS ecosystem model, and filtration by mussels (e).

**Table 1.** Components currently integrated in MOSSCO and described shortly in this paper. Several other components are under development and not listed here.

| | | |
|---|---|---|
| Pelagic ecosystem | `fabm_pelagic_component` | Sect. 3.1.1 |
| Soil ecosystem | `fabm_sediment_component` | Sect. 3.1.2 |
| 1D hydrodynamics | `gotm_component` | Sect. 3.1.3 |
| 3D hydrodynamics | `getm_component` | Sect. 3.1.4 |
| Filtration | `filtration_component` | Sect. 3.1.6 |
| Erosion/sedimentation | `erosed_component` | Sect. 3.1.5 |
| Wind waves | `simplewave_component` | Sect. 3.1.7 |
| NetCDF output | `netcdf_component` | Sect. 3.2.1 |
| NetCDF input | `netcdf_input_component` | Sect. 3.2.2 |
| Link connector | `link_connector` | Sect. 3.3.1 |
| Copy connector | `copy_connector` | Sect. 3.3.1 |
| Nudge connector | `nudge_connector` | Sect. 3.3.1 |
| Tracer transport | `transport_connector` | Sect. 3.3.2 |
| Benthic–pelagic coupling | `soil_pelagic_connector` | Sect. 3.3.3 |
| | `pelagic_soil_connector` | Sect. 3.3.3 |

**Table 2.** Acronyms and model abbreviations used in the text.

| | |
|---|---|
| bash | GNU Bourne-again shell |
| BFM | Biogeochemical Flux Model (ecosystem model) |
| BGC | Biogeochemistry |
| BMI | Basic Model Interface (coupling concept) |
| CC-by-SA | Creative Commons Attribution Share-Alike license |
| CF | NetCDF Climate and Forecast convention |
| CIM | Common Information Model (metadata standard) |
| CMI | Component Model Interface (coupling concept) |
| CMIP | Climate Model Intercomparison Project |
| COAWST | Coupled Ocean-Atmosphere-Wave-Sediment Transport |
| CSDMS | Community Surface Dynamics Modeling System |
| DKRZ | Deutsches Klimarechenzentrum (HPC center) |
| ESM | Earth System Model |
| ESMF | Earth System Modeling Framework |
| FABM | Framework for Aquatic Biogeochemical Models |
| FMS | Flexible Modeling System (coupling technology) |
| FONA | Forschung für Nachhaltigkeit (funding scheme) |
| FVCOM | Finite Volume Coastal Ocean Model |
| GCC | GNU Compiler Collection |
| GETM | General Estuarine Transport Model (3D coastal ocean model) |
| GEOS-5 | Goddard Earth Observing System version 5 |
| GNU | GNU is Not Unix |
| GOTM | General Ocean Turbulence Model (1D water column model) |
| GPL | General Public License |
| GSN | Geoscience Standard Names Ontology |
| HLRN | Norddeutscher Verbund für Hoch- und Höchstleistungsrechnen |
| HPC | High-performance computing |
| ICON | Icosahedral Non-Hydrostatic Model |
| I/O | Input and output |
| JSC | Jülich Supercomputing Centre |
| MAPL | Modeling, Analysis and Prediction Layer |
| MAECS | Model for Adaptive Ecosystems in Coastal Seas |
| MCT | Model Coupling Toolkit |
| MESSy | Modular Earth Submodel System |
| MITgcm | Massachusetts Institute of Technology Global Circulation Model |
| MOM | Modular Ocean Model |
| MOSSCO | Modular System for Shelves and Coasts |
| MPI | Message Passing Interface |
| MSPD | European Union Marine Strategic Planning Directive |
| NEMO | Nucleus for European Modeling of the Ocean |
| NetCDF | Network Common Data Form |
| NPZD | Nutrient, phytoplankton, zooplankton, detritus (ecosystem model) |
| NUOPC | National Unified Operational Prediction Capability |
| OASIS | Ocean Atmosphere Sea Ice Soil coupler |
| OMEXDIA | Ocean Margin Exchange Experiment early diagenetic model |
| OMEXDIA_P | OMEXDIA with added phosphorous |
| OMUSE_P | Oceanographic Multipurpose Software Environment |
| PARSE | Prepare, Adapt, Register, Schedule, Execute methodology |
| PISCES | Pelagic Interactions Scheme for Carbon and Ecosystem Studies |
| POM | Particulate organic matter |
| POC | Particulate organic carbon |
| RegESM | Regional Earth System Model |
| ROMS | Regional Ocean Modeling System |
| SCRIP | Spherical Coordinate Remapping and Interpolation Package |
| SNS | Southern North Sea |
| SPM | Suspended particulate matter |
| SWAN | Simulating Waves Nearshore |
| WAM | Wave Atmospheric Model |
| WFD | European Union Water Framework Directive |
| WRF | Advanced Research Weather Research and Forecasting |
| XIOS | XML Input/Output Server |
| XML | eXtensible Markup Language |
| YAML | YAML Ain't Markup Language |