# Peer review of "Modular System for Shelves and Coasts (MOSSCO v1.0) – a flexible and multi-component framework for coupled coastal ocean ecosystem modelling"

_Geoscientific Model Development, 2017_

## Author Comment (AC1) · 17 Aug 2017

The authors gratefully acknowledge the computing time granted by the John von Neumann Institute for Computing (NIC) and provided on the supercomputer JURECA at at Forschungszentrum Jülich.

---

## Referee Comment (RC1) · Anonymous Referee #1 · 23 Aug 2017

In the manuscript 'Modular System for Shelves and Coasts (MOSSCO v1.0) – a flexible and multi component framework for coupled coastal ocean ecosystem modelling' the authors present a new modelling platform enabling coupling facilities between various hydrological and biogeochemical models. The papers is well structured and clear but, in my view, it doesn't fit GMD standards for publication. In particular, the manuscript doesn't include scientific results or evaluation of the model software—at least quantitative evaluation of the modelling performance. It doesn't provide namelists or coupling procedures to support statement of modularity. Besides, It mainly relies on former

published softwares.

I do have concerns about the added value of this manuscript since most of the examples of the MOSSCO software have been detailled in independent papers and have been submitted elsewhere. Indeed most of the examples provide in the current manuscript relies on "submitted papers" or "to be submitted papers" without further details. Those examples —if detailed— might help the reader to understand how the various modules works together in sequential or parallel modes. Without those examples, it is unclear in which case or scientific questions coupled modular shelves-to-ocean models are required.

Specific comments: P5 L4Âă: For historic => historical P5 L12-15: clear a bad example because PISCES and BFM are both coupled to NEMO and other hydrodynamical models like ROMS.

---

## Author Comment (AC2) · 25 Aug 2017

We thank the Anonymous Reviewer for their comments and provide a point-by-point response below.

[Figure]

**1  GMD standards**

*The papers is well structured and clear but, in my view, it doesn't fit GMD standards for publication. In particular, the manuscript doesn't include scientific results or evaluation of the model software—at least quantitative evaluation of the modelling performance.*

We disagree. This manuscript meets the standards layed down for a *Model description paper* as outlined in the GMD manuscript types specification (https://www.geoscientific-model-development.net/about/manuscript_types.html). In particular, we "describe model components and modules, as well as frameworks and utility tools used to build practical modelling systems, such as coupling frameworks or other software toolboxes with a geoscientific application". The purpose of this manuscript is *not* a model evaluation.

**2  No namelists / Sequential vs. concurrent mode**

*It doesn't provide namelists or coupling procedures to support statement of modularity. [. . . ] examples—if detailed—might help the reader to understand how the various modules works together in sequential or parallel modes.*

The coupling mode is discussed in our subsection "Scheduling in a coupled system", and the coupling procedure is explained in textual form as well as in Figure 2. We will add in a revised version coupling namelists to exemplify the different coupling modes and add a section with a technical example of how the models work together (see below).

[Figure]

**3   Reliance on published software**

*I do have concerns about the added value of this manuscript since most of the
examples of the MOSSCO software have been detailed in independent papers and
have been submitted elsewhere. It mainly relies on former published softwares.*

It is the purpose of the MOSSCO framework to make published software interopera-
ble; thus it is desirable to rely on formerly published software. The added value is in
interoperability; indeed, we added and described for all of the published software new
coupling layers (BMI and CMI).

**4   No modeling details**

*Most of the examples provide in the current manuscript relies on submitted papers
or to be submitted papers without further details. [. . . ] Without those examples, it
is unclear in which case or scientific questions coupled modular shelves- to-ocean
models are required.*

The focus of our manuscript is the coupling concept and technical implementation,
it is not its application. The applications of the framework are detailed in separate
manuscripts; these deal with the science within and between the coupled scientific
models and contain evaluation of the scientific models.

The Southern North Sea (manuscripts submitted) examples of application are not criti-
cal for the purpose of this manuscript; they are showcases of scope and applicability of
the model framework presented. While the two manuscripts referred to have not been
published, we can provide this material confidentially on request. We will consider to
expand the 1D application (section 4.1) to include more details on the coupling (now

contained in the proceedings article by Hofmeister 2014 et al.) of the "Helgoland station" example, as this example shows most clearly the soil–pelagic coupling and the modularization process.

Thanks for this very important remark that it remained unclear when coupled modular systems are required. We argue that they are not *required* for any single application at all, one can always choose to hardwire all coupled subsystems into a monolithic code. Instead, modular models provide better potential for re-use, intercomparison, and model exchange, and thus further the advancement of a coastal geoscientific community science. We will better clarify this early on in a revised version.

Finally, we thank the reviewer for specific comments, which will all be addressed.

---

## Short Comment (SC1) · 18 Sep 2017

Dear Colleagues,

here are some short notes / clarifications about the Modular Earth Submodel System (MESSy), which you cite as (page 4, lines 4-7):

"Vice versa, a typical process coupling infrastructure like the Modular Earth Submodel System (MESSy, Jöckel et al., 2005) has been proposed to link processes across domains, but so far includes mostly atmospheric processes."

[Figure]

Since the development cycle 2 of MESSy (Jöckel et al., 2010), we do not longer distinguish between process and domain coupling from the technical perspective, but consider more the "granularity" on which model components are coupled, and whether internal or external coupling is more feasible (see Kerkweg & Jöckel (2012a), Appendix A, for a definition of coupling approaches).

A useful approach largely depends on the desired application and on the structure of the used legacy codes. If there would be no legacy code used, the finer the granularity the better it is in terms of flexibility, and this is indeed possible without deteriorating high performance computing (HPC) performance.

As documented by Pozzer et al. (2011), we use for instance the MESSy middleware for a domain coupling (atmosphere - ocean). This proves already your statement that "The differentiation between domain and process coupling is not a technical necessity". Indeed, it is not!

Moreover, as shown by Kerkweg & Jöckel (2012a,b), the MESSy middleware (with the extension of MMD, the multi-model driver) is used for on-line nesting of regional models into a global model in a MPMD (multiple program multiple data) approach. This is neither domain nor process coupling in the classical sense.

References:

Jöckel, P., Kerkweg, A., Pozzer, A., Sander, R., Tost, H., Riede, H., Baumgaertner, A., Gromov, S., & Kern, B.: Development cycle 2 of the Modular Earth Submodel System (MESSy2), Geoscientific Model Development, 3, 717–752, doi: 10.5194/gmd-3-717-2010, URL http://www.geosci-model-dev.net/3/717/2010/ (2010)

Pozzer, A., Jöckel, P., Kern, B., & Haak, H.: The Atmosphere-Ocean General Circulation Model EMAC-MPIOM, Geoscientific Model Development, 4, 771–784, doi: 10.5194/gmd-4-771-2011, URL http://www.geosci-model-dev.net/4/771/2011/ (2011)

Kerkweg, A. & Jöckel, P.: The 1-way on-line coupled atmospheric chemistry model

system MECO(n) – Part 2: On-line coupling with the Multi-Model-Driver (MMD), Geoscientific Model Development, 5, 111–128, doi: 10.5194/gmd-5-111-2012, URL http://www.geosci-model-dev.net/5/111/2012/ (2012a)

Kerkweg, A. & Jöckel, P.: The 1-way on-line coupled atmospheric chemistry model system MECO(n) – Part 1: Description of the limited-area atmospheric chemistry model COSMO/MESSy, Geoscientific Model Development, 5, 87–110, doi: 10.5194/gmd-5-87-2012, URL http://www.geosci-model-dev.net/5/87/2012/ (2012b)

---

## Referee Comment (RC2) · Anonymous Referee #2 · 28 Sep 2017

**General Comments**

The manuscript introduces novel work about interoperability of the existing coupling libraries/technologies such as ESMF, BMI to address multi-scale, multi-dimension coupling of shelf and coastal sea processes. The manuscript is also extended to include the brief results of newly designed modeling system (MOSSCO) by providing different combinations of the component of modeling system to solve different problems, which are described detailed in referanced papers. The flexibility of the modeling system also

allows to define variety of applications in a range of one-dimensioal models to three-dimensional hydrodinamical model components, which is also important contribution of the current work even still performance of the modelling system is questionalble. Some details about coupling of ESMF and BMI is still missing and also it would be usefull to include some performance benchmark of different model configurations.

**Specific Comments**

- In Introduction section (probably in the section after line 25-30 in page 2), following items can be given as other efforts to create modular and generic modeling systems/frameworks

  **OMUSE framework**

  – https://www.projects.science.uu.nl/oceanclimate/nonlin_dynamics.php2-4
  – https://imum2016.sciencesconf.org/data/pages/pelupessy.pdf

  **CSDMS (community surface dynamics modeling system)**

  – http://csdms.colorado.edu/wiki/Main_Page

- **page 5, line 31:** what type of coupling used in sequential mode. explicit or semi-implicit? it is better to be more clear about the execution order of the model components and supported coupling time step configurations. most of the model components (expecially 1d models) that are used in the framework are sequential codes. To that end, how this is handled with the modeling framework in terms of efficency and scalablity of the overall system when 3d ocean model compoenent and multiple 1d and 2d model components are used together. what will be the distribution of the model components across the underlying hardware system?

[Figure]

it is better to give more information about the overall overhead of the coupling interface and pottential bottlenecks.

- **section 2.3:** it will be better to include simple text based specification to create ESMF driver component along with the command to trigger it. It is also same for the command line utility. Also, it could be good to mention similarity and difference with Cupid Eclipse interface (https://www.earthsystemcog.org/projects/cupid/) about creating driver component automatically.

- How sediment component inherits grid information from the coupled system. Is NUOPC grid transfer feature used in here? It is better to add detail about the design. Again, how GETM export its grid infotmation to other components (section 3.1.4).

- **section 3.3.2:** Is this component solves transport equations on the fly using input data coming from 3d ocean model component using mediator component? or it just exports data to ocean component to calculate transport inside of it. please clarify it.

- **section 5:** conservation of mass and energy fluxes plays important role in the overall perfromance of the modelling system and needs to be addresses carrefully. It is better to extend discussion about conservation of mass and energy fluxes. How MOSSCO handles conservation expecially in the context of coupling in different dimensionality such as one and three dimensional models? The ESMF library supports only first order conservation, which does not perform well expecially when the resolution of computational grid of different model components is high.

**Technical Corrections**

- **page 7, line 21:** replace "meta data" with "metadata"

- **page 8, line 6:** replace "meta data" with "metadata"

- **page 10, line 13:** replace "meta data" with "metadata"

- **page 13, line 18:** (to be submitted) ? it is better to give it in the reference section

- **page 13, line 28:** (submitted) need to be replaced with a appropriate reference by indicating that it is still in under review

- **page 15, line 6:** remove ? or replace with /

- **page 15, line 14 and 15:** couplable? Couplability? it is better to rephase the sentence.

- **page 16, line 17:** replace : with .

- **page 17, line 9:** The reference given in the manuscript is belongs to workflow integrated atmosphere-ocean modeling system (WRF+ROMS), which is the early attempt to create coupled modeling system with semi-autonomous configuration and setup enwironment. The correct reference for RegESM modeling system can be found in following links

  - https://link.springer.com/article/10.1007/s00382-016-3241-1
  - for wave coupling http://meetingorganizer.copernicus.org/EGU2015/EGU2015-3644.pdf
  - The two pomponent prototype version is in https://www.geosci-model-dev.net/6/283/2013/

---

## Author Comment (AC3) · 27 Nov 2017

We thank Dr. Jöckel for bringing to our attention the recent developments in the Modular Earth Submodel System (MESSy Jöckel et al. 2005).

We stated in our manuscript that "a typical process coupling infrastructure like the . . . MESSy . . . so far includes mostly atmospheric processes". Dr. Jöckel's comment advises us that since the development cycle 2 of MESSy (Jöckel et al., 2010), they "do not longer distinguish between process and domain coupling from the technical per-

spective, but consider more the granularity on which model components are coupled".

As Dr. Jöckel notes, this cycle 2 development is in line with our statement "The differentiation between domain and process coupling is not a technical necessity"; in an updated version of the manuscript we will be happy to reference the relevant MESSy developments and their consideration of granularity rather than domain vs. process coupling; their prior work (e.g. Kerkweg and Jöckel, 2012) substantiates the MOSSCO coupling approach that also does not differentiate per se between process and domain coupling.

**References**

Jöckel, P., Sander, R., Kerkweg, A., Tost, H., and Lelieveld, J.: Technical Note: The Modular Earth Submodel System (MESSy) - a new approach towards Earth System Modeling, Atmospheric Chemistry and Physics, 5, 433–444, doi:10.5194/acp-5-433-2005, 2005.

Jöckel, P., Kerkweg, A., Pozzer, A., Sander, R., Tost, H., Riede, H., Baumgaertner, A., Gromov, S., and Kern, B.: Development cycle 2 of the Modular Earth Submodel System (MESSy2), Geoscientific Model Development, 3, 717–752, doi:10.5194/gmd-3-717-2010, 2010.

Kerkweg, A. and Jöckel, P.: The 1-way on-line coupled atmospheric chemistry model system MECO(n) – Part 2: On-line coupling with the Multi-Model-Driver (MMD), Geoscientific Model Development, 5, 111–128, doi:10.5194/gmd-5-111-2012, 2012.

---

## Author Response (AR1)

**Point-by-point reply to issues raised by reviewers and commentators**

C. Lemmen et al.

November 27, 2017

We thank all reviewers and discussion participants for comments raised during the discussion period of our manuscript. Below, we address each of the issues raised.

**Short comment by Dr. Jöckel**

We stated in our manuscript that "a typical process coupling infrastructure like the .. MESSy .. so far includes mostly atmospheric processes". Dr. Jöckel's comment advises us that since the development cycle 2 of MESSy (Jöckel et al., 2010) they

1. ... *do not longer distinguish between process and domain coupling from the technical perspective, but consider more the granularity on which model components are coupled*

   **Response:** As Dr. Jöckel notes, this cycle 2 development is in line with our statement "The differentiation between domain and process coupling is not a technical necessity"; we now write "Vice versa, a typical process coupling infrastructure like the Modular Earth Submodel System (MESSy, Jöckel et al., 2005), which initially linked mostly atmospheric processes, has been generalized to support linking at a user-chosen granularity irrespective of the process versus domain dichotomy (e.g., Kerkweg and Jöckel, 2012)."

**1 Anonymous Reviewer 1**

1. *The papers is well structured and clear but, in my view, it doesn't fit GMD standards for publication. In particular, the manuscript doesn't include scientific results or evaluation of the model software – at least quantitative evaluation of the modelling performance.*

   **Response:** We disagree. This manuscript meets the standards layed down for a *Model description paper* as outlined in the GMD manuscript types specification (https://www.geoscientific-model-development.net/about/manuscript_types.html). In particular, we "describe model components and modules, as well as frameworks

and utility tools used to build practical modelling systems, such as coupling frameworks or other software toolboxes with a geoscientific application". The purpose of this manuscript is *not* a model evaluation.

2. *It doesn't provide namelists or coupling procedures to support statement of modularity. ...examples – if detailed – might help the reader to understand how the various modules works together in sequential or parallel modes.*

   **Response:** The coupling mode is discussed in our subsection "Scheduling in a coupled system", and the coupling procedure is explained in textual form as well as in Figures 2 and 3. Based on suggestions also by reviewer 2, we added an example of a coupling namelist (new Figure 3) to make this more explicit.

   **Response:** We also added a workflow description of how (in an example) a coupled model systems transfers information to the application section

3. *I do have concerns about the added value of this manuscript since most of the examples of the MOSSCO software have been detailed in independent papers and have been submitted elsewhere. It mainly relies on former published softwares.*

   **Response:** It is the *purpose* of the MOSSCO framework to make published software interoperable; thus it is desirable to rely on formerly published software. The added value is in interoperability; indeed, we added and described for all of the published software new coupling layers (BMI and CMI).

4. *Most of the examples provide in the current manuscript relies on submitted papers or to be submitted papers without further details. .. Without those examples, it is unclear in which case or scientific questions coupled modular shelves- to-ocean models are required.*

   **Response:** The focus of our manuscript is the coupling concept and technical implementation, it is not its application. The applications of the framework are detailed in separate manuscripts; these deal with the science within and between the coupled scientific models and contain evaluation of the scientific models. Meanwhile, the manuscript by Kerimoglu et al. (2017) has been published, the paper by Nasermoaddeli et al. (2017) has been accepted and is "in press", and the paper by Slavik et al. (2017) is under review; it is available for inclusion in the review process from the arXiv.

   **Response:** It is a very important remark that it remains unclear when coupled modular systems are required. We argue that they are not *required* for any single application at all, one can always choose to hardwire all coupled subsystems into a monolithic code. Instead, modular models provide better potential for re-use, intercomparison, and model exchange, and thus further the advancement of a coastal geoscientific community science. We have clarified this in the discussion.

**Anonymous Reviewer 2**

1. *...performance of the modelling system is questionable. Some details about coupling of ESMF and BMI is still missing and also it would be useful to include some performance benchmark of different model configurations.*

    **Response:** As the coupling framework is flexible, the performance of the system depends on the coupled components and on the performance of the infrastructure used. A typical speedup achieved, e.g., in the bivalve application on 192 processors is 3000: a full simulation year is completed in three compute-time hours, such a speedup allows decadal up to multidecadal simulations. One of the identified bottlenecks (that varies strongly with the HPC system used) is data transfer from memory to disk: this will be in the future addressed by the use of parallel NetCDF and/or leveraging the XML I/O Server.

    **Response:** We ensured that it is now clearer that we use BMI conceptually and ESMF/NUOPC phases in the actual implementation.

2. *...following items can be given as other efforts to create modular and generic modeling systems/frameworks: OMUSE, CSDMS.*

    **Response:** We added references and discussion of these two frameworks to the introduction and to the design section.

3. *...what type of coupling used in sequential mode. explicit or semi-implicit? it is better to be more clear about the execution order of the model components and supported coupling time step configurations. most of the model components (expecially 1d models) that are used in the framework are sequential codes. To that end, how this is handled with the modeling framework in terms of efficency and scalablity of the overall system when 3d ocean model compoenent and multiple 1d and 2d model components are used together. what will be the distribution of the model components across the underlying hardware system? it is better to give more information about the overall overhead of the coupling interface and potential bottlenecks.*

    **Response:** We added more information on the coupling in sequential mode. There is no coupling between 1D and 3D components in the examples that we currently operate. All of the current examples run in sequential mode on the same compute element distribution. A regridding implementation, also with redistribution between different compute elements is currently in development and will be described in a different manuscript. We added performance, coupling overhead and bottlenecks in the discussion section.

4. *...it will be better to include simple text based specification to create ESMF driver component along with the command to trigger it. It is also same for the command line utility. Also, it could be good to mention similarity and difference with Cupid Eclipse interface (https://www.earthsystemcog.org/projects/cupid/) about creating driver component automatically.*

**Response:** We added a new figure showcasing a coupling configuration. The section on driver component creation was expanded according to the above comments.

5. *How sediment component inherits grid information from the coupled system. Is NUOPC grid transfer feature used in here? It is better to add detail about the design. Again, how GETM export its grid information to other components (section 3.1.4).*

   **Response:** We expanded the section describing grid inheritance. We also added a subsection in the applications section that exemplifies the data flow in a modularly coupled application.

6. *Is this component solves transport equations on the fly using input data coming from 3d ocean model component using mediator component? or it just exports data to ocean component to calculate transport inside of it. please clarify it.*

   **Response:** We clarified this in the description of the transport connecter. Indeed, the transport connector only repackages the tracer fields such that they can be transported within the hydrodynamic component.

7. *. . . conservation of mass and energy fluxes plays important role in the overall performance of the modelling system and needs to be addresses carefully. It is better to extend discussion about conservation of mass and energy fluxes. How MOSSCO handles conservation expecially in the context of coupling in different dimensionality such as one and three dimensional models? The ESMF library supports only first order conservation, which does not perform well expecially when the resolution of computational grid of different model components is high.*

   **Response:** We added a cautionary note on potential divergence of mass introduced by the coupling framework itself in the section describing sequential coupling. As the implementation of specific scientific couplers that, e.g. calculate ocean–atmosphere fluxes, is beyond the scope of the coupling framework itself, we added consideration of conservation in the sections describing special currently existing couplers where conservation is indeed challenging.

8. *Technical Corrections*

   **Response:** We addressed all technical corrections

[revised manuscript text omitted]

---

## Editor Decision (ED1)

Dear authors,

Thank you for your revised manuscript that addresses many of the Reviewers' comments. However, while I don't share Reviewer 1's concern about the fact that your manuscript would not fit GMD standards for publication, I think that you don't fully address Reviewer 2' more concrete comments.

My main concern rejoins Reviewer 2' comments #1, #3, and #5 about the resulting layout of the components of a MOSSCO application on the available computing ressources and the related question of performance. While I fully understand that each application will be specific and that no general assertion can be formulated, I still think that these aspects need clarification in the manuscript. In particular :

- New section 4.4 : Thanks for adding this section but it is quite difficult to follow without an illustration. Please add a figure sketching the layout of the components on the computing resources used and the coupling interactions between them.
- p.6, l.15 to 22 : Thanks for adding this paragraph but I am still not sure which component layout would result from the notation in Fig 3b. As B runs after A and C after B, does it necessarily mean they run sequentially on the same set of computing resources? Or could it be that A and B have each their own set of computing ressources, so that A runs for the first coupling period on its resources, then B runs for that first coupling period on its resources with A also running concurrently on its ressources for the next coupling period?
- p.6, l.13 : why do you qualify of "sequential" the coupling of Fig 2; in this example
- the components A, B and C obviously run concurrently.
- p.17, last paragraph : This assertion on scalability is not meaningful; you have to state that the scalability of MOSSCO applications directly depends on the scalability of the components and on the fact that the coupling workflow does not introduce bottleneck. The statement on the ESMF overhead is OK although a little too qualitative.
- p.18, first paragraph : the added sentences on speedup are not meaningful either ; this is a speedup with compare to what; it looks like you say that the elapse time is 3000 smaller when running on 192 processors instead of running on 1 processor but I strongly doubt this is the case.

Other comments :

- p.6, l.7 : I think this sentence is not right ; in Balaji (2016), « the atmospheric radiative transfer component has been configured to run in parallel with a composite component consisting of every other atmospheric component, including the atmospheric dynamics and all other atmospheric physics components. », please correct.
- p.7, last paragraph : What does "scaling" means in this context? Is it just that MOSCCO can couple 0-, 1-, 2-, 3-D models? If so I am not sure that "scaling" is the right word to use.
- p.11, l.30 : I would not write « identical to the output component » because the output component, when used in a parallel model, produces multiple partial files; here the input component will read the different elements in parallel from one global file.
- p.13, l.22, I think you cannot write « ranging from one-dimensional water-column to three-dimensional ... » while in your reply to reviewer 2, you write "There is no coupling between 1D and 3D components in the examples that we currently operate. »
- p.17, l.14&15 : This is misleading ; mediator components will of course have to ensure mass and energy flux when they will include regridding

Other minor comments :

- p.3, l.1 : for OASIS reference, please use : A. Craig, S. Valcke, L. Coquart, 2017: Development and performance of a new version of the OASIS coupler, OASIS3-MCT_3.0, Geosci. Model Dev., 10, 3297-3308, https://doi.org/10.5194/gmd-10-3297-2017, 2017.

- p.3, l.2 : for ESMF reference, I think it is better to use : Theurich, G., Deluca, C., Campbell, T., Liu, F., Saint, K., Verten- stein, M., Chen, J., Oehmke, R., Doyle, J., Whitcomb, T., Wall- craft, A., Iredell, M., Black, T., Da Silva, A. M., Clune, T., Fer- raro, R., Li, P., Kelley, M., Aleinov, I., Balaji, V., Zadeh, N., Ja- cob, R., Kirtman, B., Giraldo, F., McCarren, D., Sandgathe, S., Peckham, S., and Dunlap IV, R.: The Earth System Prediction Suite: Toward a Coordinated U.S. Modeling Capability, B. Am. Meteor. Soc., 97, 1229–1247, https://doi.org/10.1175/BAMS-D- 14-00164.1, 2016.
- p.3, l.5 : What do you mean by "very differently represented"?
- p.3, l.18 : Given MOSSCO, I think you should not start by « Currently, there is no ... » ; maybe replace this by something like « Up to now, there was no ... »
- p.3, l.29 : This sentence is a bit too heavy, please rephrase it ; I don't think that the design of a software should emphasize the needs of researchers, instead it should answer them !
- p. 5, l.5 : I am not sure « coupleability » is a real English word ; maybe put this word between quotes, at least the first time it appears in the text.
- p.5, l.7 : maybe change « For the run phase, it is mandatory that this phase refers to » for « For the run phase, it is mandatory to refer to »
- p.5, l.13 : why is « operation » between brackets ? I think this level of (unclear) detail is not needed here.
- p.6, l.15 : I don't understand what « recursive acronym YAML Ain't Markup Language » means.
- p.15, l.24 : « … components. In the ...» instead of « … components. in the ...»
- p.16, l.17 : Please remove « indeed » or move it at the beginning of the sentence.
- p.18, l.6 : This sentence is awkward ; maybe replace it by a simpler sentence like : « Multi-component systems may also suffer from low acceptance by the research community. »
- p.18, l.13 : I am not sure what « up-scaling » means in this context.
- p.18, l.24 : « benefit from each others' progress » ; is the English correct ? Maybe replace by « benefit from each other's progress » or « benefit from all others' progress » ?
- p.19, l.33 : please replace OASIS/MCT by OASIS3-MCT

With best regards,
 Sophie Valcke

---

## Author Response (AR2)

**Point-by-point reply to final comments by editor requesting minor changes GMD-2017-138**

**C. Lemmen et al.**

**January 12, 2018**

Dear Dr. Valcke,

thank you for your additional efforts to improve our paper. We are very grateful for your diligent second check of the manuscript and for identifying few remaining issues. These are now fully addressed in our revision. Please find a point-by-point reply below.

1. *New section 4.4 : Thanks for adding this section but it is quite difficult to follow without an illustration. Please add a Figure sketching the layout of the components on the computing resources used and the coupling interactions between them.*

   **Response:** All components in this example run sequentially on the same resources, thus a sketch of the distribution on compute resources would not help. We made this clear in the text by adding "This coupled application is to be deployed in sequential mode on the same set of compute resources for all 13 components". Nonetheless we appreciate your confusion (see also your issues below) and agree to make this point much more clear. We have given special attention throughout the text for potential pitfalls using the words "shared" and rephrased sentences that involve sequential/concurrent coupling statements. We also elaborated the workflow section in several places to explain how this concrete example is related to the coupling configuration template (Figure 3).

2. *p.6, l.15 to 22 : Thanks for adding this paragraph but I am still not sure which component layout would result from the notation in Fig 3b. As B runs after A and C after B, does it necessarily mean they run sequentially on the same set of computing resources? Or could it be that A and B have each their own set of computing resources, so that A runs for the first coupling period on its resources, then B runs for that first coupling period on its resources with A also running concurrently on its resources for the next coupling period?*

   **Response:** We clarified this section and related it better to Figure 3. We also addressed the confusion between sequential and concurrent coupling. We added "all components run on shared resources in (default) sequential coupling mode."

3. *p.6, l.13 : why do you qualify of "sequential" the coupling of Fig 2; in this example the components A, B and C obviously run concurrently.*

   **Response:** They *seemingly* run concurrently, but this is an incorrect perspective. The time axis is in simulation time, and all components share (in "parallel") the simulation time. In a wall-clock perspective, they run concurrently. We added the statement "shown along a simulation time axis, which is independent of the type of (sequential or concurrent) deployment" in Fig. 2's caption.

4. *p.17, last paragraph : This assertion on scalability is not meaningful; you have to state that the scalability of MOSSCO applications directly depends on the scalability of the components and on the fact that the coupling workflow does not introduce bottleneck. The statement on the ESMF overhead is OK although a little too qualitative.*

   **Response:** We revised the paragraph in multiple places, avoided the confusing term speedup and provided a percentage ESMF overhead from our scaling experiments.

5. *p.18, first paragraph: the added sentences on speedup are not meaningful either ; this is a speedup with compare to what; it looks like you say that the elapse time is 3000 smaller when running on 192 processors instead of running on 1 processor but I strongly doubt this is the case.*

   **Response:** Please see above

6. *p.6, l.7 : I think this sentence is not right ; in Balaji (2016), the atmospheric radiative transfer component has been conFigured to run in parallel with a composite component consisting of every other atmospheric component, including the atmospheric dynamics and all other atmospheric physics components. , please correct.*

   **Response:** We meant the same but you put it better. We now rephrased to "In their system, an ocean and an atmosphere component run concurrently; within the atmospheric component, the radiation code is executed concurrently to a composite component containing, in turn, a sequential coupling of all non-radiative atmospheric processes."

7. *p.7, last paragraph : What does "scaling" means in this context? Is it just that MOSCCO can couple 0-, 1-, 2-, 3-D models? If so I am not sure that scaling is the right word to use.*

   **Response:** We avoided using scaling (or upscaling) in this context as suggested. We clarified "As a concrete example, the novel Model for Adaptive Ecosystems in Coastal Seas (MAECS) has been developed iterating between an application for the lab (zero-dimensional) scale and the three-dimensional regional coastal ocean scale."

8. *p.11, l.30 : I would not write identical to the output component because the output component, when used in a parallel model, produces multiple partial files; here the input component will read the different elements in parallel from one global file.*

**Response:** In fact, the input can come from distributed files as well as a global file. We removed the word parallel to avoid confusion, we rephrased "It inherits its decomposition from other components in the coupled system. Data can be read from a single file for the entire domain or from distributed files for all decomposed compute elements separately. "

9. *p.13, l.22, I think you cannot write  ranging from one-dimensional water-column to three- dimensional …  while in your reply to reviewer 2, you write ?There is no coupling between 1D and 3D components in the examples that we currently operate.*

**Response:** We removed the confusing statement and merely talk about applications at different stations, transects, and sea domains.

10. *p.17, l.14&15 : This is misleading ; mediator components will of course have to ensure mass and energy flux when they will include regridding*

**Response:**  We rephrased the paragraph to address this concern.

11. *p.3, l.1 : for OASIS reference, please use: Craig et al. 2017.*
    *p.3, l.2 : for ESMF reference, I think it is better to use : Theurich et al. 2016*

**Response:** We changed the citations according to the above suggestions

12. *p.3, l.5 : What do you mean by "very differently represented"*

**Response:** rephrased and elaborated as "Their intention is to provide a high-level user interface and infrastructure for coupling existing and new oceanographic models whose spatial representation differs greatly, in particular between Lagrangian and Eulerian representations."

13. *p.3, l.18 : Given MOSSCO, I think you should not start by  Currently, there is no …   ; maybe replace this by something like  Up to now, there was no …*
    *p.3, l.29 : This sentence is a bit too heavy, please rephrase it ; I don't think that the design of a software should emphasize the needs of researchers, instead it should answer them !*

**Response:** We changed the text accordingly.

14. *p. 5, l.5 : I am not sure  coupleability  is a real English word ; maybe put this word between quotes, at least the first time it appears in the text.*

**Response:** rephrased as "and provides the basic coupling ability—or "coupleability"— of the model"

15. *p.5, l.7 : maybe change  For the run phase, it is mandatory that this phase refers to  for  For the run phase, it is mandatory to refer to*
    *p.5, l.13 : why is  operation  between brackets ? I think this level of (unclear) detail is not needed here.*

**Response:** We changed the text accordingly.

16.  *p.6, l.15 : I don't understand what  recursive acronym YAML Ain't Markup Language  means.*

**Response:** There's not much we can do about that. It is meaningless, just like GNU expands to GNU is Not Unix. We rephrased and added a link: "Users specify the coupling in a text format using YAML (short for YAML Ain't Markup Language, http://yaml.org) notation,"

17.  *p.15, l.24 :  ... components. In the ... instead of  ... components. in the ...*
*p.16, l.17 : Please remove  indeed  or move it at the beginning of the sentence.*
*p.18, l.6 :  This sentence is awkward ; maybe replace it by a simpler sentence like :   Multi-component systems may also suffer from low acceptance by the research community.*

**Response:** We changed the text according to the above suggestions

18. *p.18, l.13 : I am not sure what  up-scaling  means in this context.*

**Response:**  We rephrased: "Yet, it is not clear whether the bottom-up approach of many interacting modular components leads to an emergent system behaviour that is desirable and exhibits new insights or whether the system gets tangled up in coupling complexity."

19.  *p.18, l.24 :   benefit from each others? progress  ; is the English correct ?  Maybe replace by benefit from each other's progress   or   benefit from all others' progress   ?  p.19, l.33 : please replace OASIS/MCT by OASIS3-MCT*

**Response:** We changed the text according to the above suggestions

Sincerely
Carsten Lemmen (on behalf of all authors)

[revised manuscript text omitted]